# A direct black-hole mass measurement in a little red dot at high redshift

Ignas Juodžbalis[1,2 ✉], Cosimo Marconcini[3,4], Francesco D'Eugenio[1,2], Roberto Maiolino[1,2,5], Alessandro Marconi[3,4], Hannah Übler[6], Jan Scholtz[1,2], Xihan Ji[1,2], Gareth C. Jones[1,2], Michele Perna[7], Santiago Arribas[7], Jake S. Bennett[8], Volker Bromm[9], Andrew J. Bunker[10], Stefano Carniani[11], Stéphane Charlot[12], Giovanni Cresci[4], Pratika Dayal[13,14], Eiichi Egami[15], Andrew Fabian[16], Kohei Inayoshi[17], Yuki Isobe[1,2,18], Lucy R. Ivey[1,2], Sophie Koudmani[19,20], Nicolas Laporte[21], Boyuan Liu[22], Jianwei Lyu[15], Giovanni Mazzolari[6], Stephanie Monty[16], Eleonora Parlanti[11], Pablo G. Pérez-González[7], Brant Robertson[23], Raffaella Schneider[24], Debora Sijacki[1,16], Sandro Tacchella[1,2], Alessandro Trinca[25], Rosa Valiante[26], Marta Volonteri[12], Joris Witstok[27,28] & Saiyang Zhang[29,30]

Recent discoveries of faint active galactic nuclei (AGN) at the redshift frontier have revealed a plethora of broad Hα emitters with optically red continua, named little red dots (LRDs)[1], which comprise 15–30% of the high-redshift broad-line AGN population[2]. Owing to their peculiar properties[3–6], modelling LRDs with standard AGN scenarios has proven challenging. In particular, the validity of single-epoch virial mass estimates in determining the black-hole masses of LRDs has been called into question, with some models claiming that masses might be overestimated by up to two orders of magnitude[7–10]. Here we report a direct, dynamical black-hole mass measurement in a strongly lensed LRD at a redshift of 7.04. The combination of lensing with deep spectroscopic data reveals a rotation curve that is inconsistent with a nuclear star cluster, yet can be well explained by Keplerian rotation around a point mass of 50 million solar masses, consistent with virial black-hole mass estimates. The Keplerian rotation leaves little room for any stellar component in a host galaxy, as we conservatively infer $M_{BH}/M_* > 2$ (where $M_{BH}$ is the black-hole mass and $M_*$ is the stellar mass). Such a 'naked' black hole, together with its near-pristine environment[11], indicates that this LRD is a massive black-hole seed caught in its earliest accretion phase.

Abell 2744−QSO1 (hereafter QSO1) is a strongly lensed, triply imaged system, first discovered in the James Webb Space Telescope (JWST) Near Infrared Camera imaging by ref. 12, whose redshift was confirmed to be $z = 7.04$ through Near Infrared Spectrograph (NIRSpec) prism spectroscopy[13], which also revealed broad components in Hα and Hβ lines. The compactness and a red optical (rest frame) slope together with a blue ultraviolet (rest frame) slope classify it as a typical 'little red dot' (LRD)[2,14]. Further observations clearly spectrally resolved the broad- and narrow-line emission, and also detected line variability[4,15,16], thereby robustly identifying QSO1 as hosting an accreting black hole (BH). On the basis of virial relations using broad-line widths and luminosities, a BH mass of about $4 \times 10^7$ $M_\odot$ was estimated[4,13,15]. However, these results rest on the assumption that 'virial relations'[17] that are calibrated locally, are still applicable at $z = 7$. In this work, we provide a direct BH mass measurement in the high-redshift Universe, indeed illustrating that virial BH mass calibrations apply to this prototypical LRD.

It is first interesting to note that, given the low narrow-line velocity dispersion in QSO1[15] ($\sigma_N < 22$ km s$^{-1}$; Supplementary Information

[1]Kavli Institute for Cosmology, University of Cambridge, Cambridge, UK. [2]Cavendish Laboratory, University of Cambridge, Cambridge, UK. [3]Università di Firenze, Dipartimento di Fisica e Astronomia, Florence, Italy. [4]INAF - Arcetri Astrophysical Observatory, Florence, Italy. [5]Department of Physics and Astronomy, University College London, London, UK. [6]Max-Planck-Institut für extraterrestrische Physik, Garching, Germany. [7]Centro de Astrobiología (CAB), CSIC-INTA, Madrid, Spain. [8]Center for Astrophysics | Harvard & Smithsonian, Cambridge, MA, USA. [9]Department of Astronomy, University of Texas at Austin, Austin, TX, USA. [10]Department of Physics, University of Oxford, Oxford, UK. [11]Scuola Normale Superiore, Pisa, Italy. [12]Sorbonne Université, CNRS, Institut d'Astrophysique de Paris, Paris, France. [13]Kapteyn Astronomical Institute, University of Groningen, Groningen, The Netherlands. [14]Canadian Institute for Theoretical Astrophysics, University of Toronto, Toronto, Ontario, Canada. [15]Steward Observatory, University of Arizona, Tucson, AZ, USA. [16]Institute of Astronomy, University of Cambridge, Cambridge, UK. [17]Kavli Institute for Astronomy and Astrophysics, Peking University, Beijing, China. [18]Waseda Research Institute for Science and Engineering, Faculty of Science and Engineering, Waseda University, Tokyo, Japan. [19]Centre for Astrophysics Research, Department of Physics, Astronomy and Mathematics, University of Hertfordshire, Hatfield, UK. [20]St Catharine's College, University of Cambridge, Cambridge, UK. [21]Aix Marseille Université, CNRS, CNES, LAM (Laboratoire d'Astrophysique de Marseille), Marseille, France. [22]Universität Heidelberg, Zentrum für Astronomie, Institut für Theoretische Astrophysik, Heidelberg, Germany. [23]Department of Astronomy and Astrophysics, University of California, Santa Cruz, Santa Cruz, CA, USA. [24]Dipartimento di Fisica, 'Sapienza' Università di Roma, Rome, Italy. [25]Como Lake Center for Astrophysics, DiSAT, Università degli Studi dell'Insubria, Como, Italy. [26]INAF/Osservatorio Astronomico di Roma, Monte Porzio Catone, Italy. [27]Cosmic Dawn Center (DAWN), Copenhagen, Denmark. [28]Niels Bohr Institute, University of Copenhagen, Copenhagen, Denmark. [29]Department of Physics, University of Texas at Austin, Austin, TX, USA. [30]Weinberg Institute for Theoretical Physics, Texas Center for Cosmology and Astroparticle Physics, University of Texas at Austin, Austin, TX, USA. ✉e-mail: ij284@cam.ac.uk

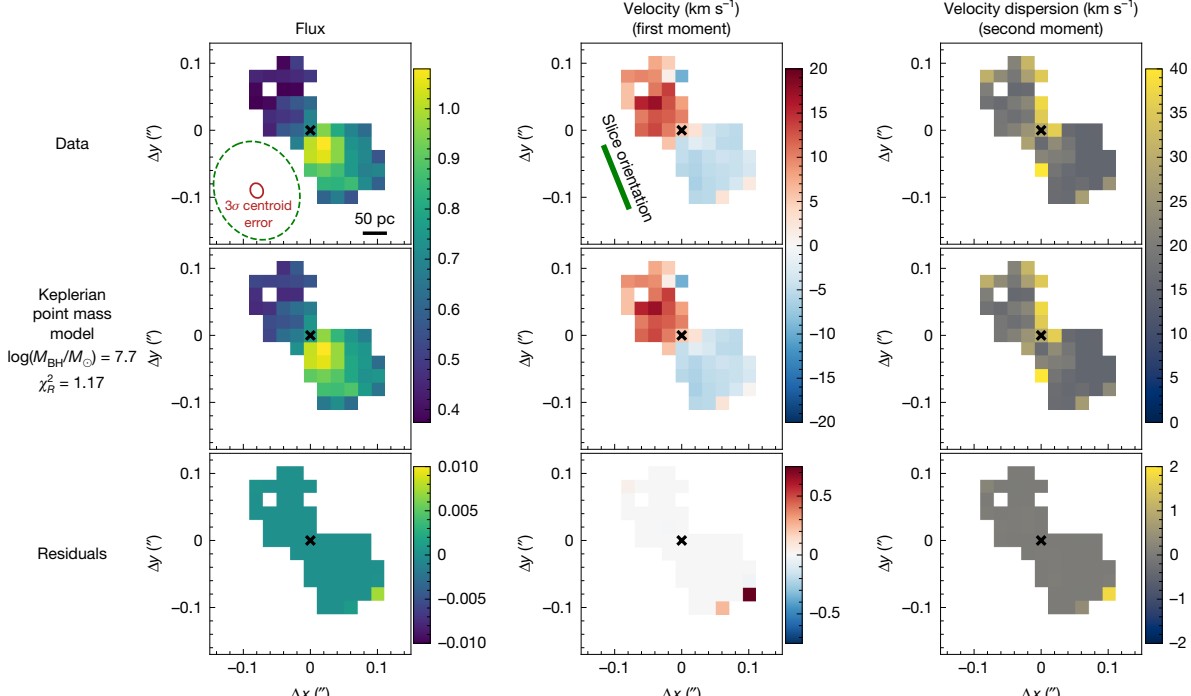

**Fig. 1 | Hα narrow-line emission and moment maps of QSO1.** The top row shows the observed integrated narrow Hα flux, and the first and second moments of the flux distribution, which correspond to line-of-sight velocity and velocity dispersion, respectively. The middle row shows the best-fit Keplerian MOKA3D model to the Hα maps. The bottom row shows the residuals of the best-fit model. The dashed green ellipse in the top-left panel shows the point spread function and the inner red solid ellipse shows $3\sigma$ uncertainty of the centroids of the velocity channels used for spectroastrometry. The green segment shows the orientation of the integral-field-unit slices, approximatively parallel to the rotation field, indicating that the latter cannot be a calibration artefact. The spatial scale is given by the black line in the top-left panel.

section 1.1), the BH's sphere of influence, assuming the mass estimated from the virial relations, has a radius of ≳270 pc, which should be resolvable with JWST, owing to the gravitational lens shearing of about 3.5 (ref. 13). However, we will not make any a priori assumptions about the BH mass in the following analysis.

We perform a detailed analysis of the spatial and kinematic information present in the current deep, high-resolution NIRSpec integral field spectroscopy data, by focusing on the narrow Hα emission. The line flux, line-of-sight velocity (first moment) and velocity dispersion (second moment) maps are shown in Fig. 1 (top). Clearly, the narrow Hα velocity field reveals a velocity gradient with an amplitude of approximately 10 km s$^{-1}$.

The narrow-line emission is spatially extended on scales of up to approximately 200 pc, as found by ref. 11 (whereas the continuum is unresolved[13]). We sample the rotation curve across 2 spatial bins corresponding to 100-pc and 150-pc distances from the rotation centre (Methods). We do not bin velocities at distances closer to the centre, as these are affected by beam smearing with the velocities on the opposite side, and require other techniques to be resolved (spectroastrometry and full three-dimensional (3D) analysis, discussed below). The binned velocities are shown as blue circles in Fig. 2.

The small size of the object makes it difficult to trace the inner parts of the rotation curve. However, given the high signal-to-noise on Hα, it is possible to recover kinematic information below the spatial resolution beam through spectroastrometric techniques. In brief, spectroastrometry consists in identifying the centroid positions when scanning spectroscopic channels of a line[18], producing a map of average gas positions across a velocity range (Extended Data Fig. 1 and Supplementary Information section 1.2). We find that the centroids of the Hα emission in the +50 km s$^{-1}$ and −50 km s$^{-1}$ velocity channels are separated by 24.9 ± 9.4 pc in the source plane (magenta crosses in Fig. 2). Taking half of this separation gives a radius scale $r_{\mathrm{spec}} = 12.5^{+4.7}_{-4.7}$ pc,

which, coupled with the velocity, yields a spectroastrometric enclosed mass in solar masses ($M_{\mathrm{spec}}/M_\odot$) of $\log(M_{\mathrm{spec}}/M_\odot) = 6.70$–7.2 (Supplementary Information section 1.2). This is a lower limit, as the inclination is unconstrained with this method. We have also leveraged spectroastrometry by splitting the line in finer velocity bins (lighter violet markers in Fig. 2; Supplementary Information section 1.2)—in this case, the signal-to-noise is lower, hence these points are less constraining, but they clearly show evidence of a rapidly declining rotation curve.

The spectroastrometric measurements above indicate a compact and dense system. However, they alone cannot exclude a significant contribution of stars, gas or dark matter to the mass budget. We therefore combine the spectroastrometric measurements with the large-scale rotation to construct a rotation curve for QSO1 (Fig. 2). We fit the data in Fig. 2 with (1) a point mass and (2) a compact, yet extended, mass distribution mimicking the (thoroughly studied) nuclear star cluster (NSC) of the Milky Way[19]. Details are given in Methods; however, Fig. 2 shows that an extended Milky-Way-like NSC mass distribution is disfavoured (reduced chi-squared $\chi^2_{\mathrm{R}} = 3.2$) when compared with a pure point mass (Keplerian) model ($\chi^2_{\mathrm{R}} = 1.0$), with BH mass, $\log(M_{\mathrm{BH}}/M_\odot) = 6.75 \pm 0.15$. The evidence from kinematics alone corresponds to >5$\sigma$ preference for a point mass. Additional kinematic evidence ruling out the Milky Way NSC (and more extensive NSC cases) are presented further below and in Methods, also through the full 3D kinematic fitting. Here we note that the implied NSC masses, $\log M_*(<R_c) = 6.52$ and $\log M_*(<100 \text{ pc}) = 7.1$, are considerably above the stellar-mass limits derived from the mass-to-light ratio of the object (Supplementary Information section 1.8). In these tests, the NSC effective radius $R_c$ was fixed to 5 pc, as in the Milky Way's NSC—if left free, the $R_c$ collapses to $10^{-4}$ pc, effectively mimicking a point-mass. In Methods, we estimate an upper limit on $R_c$ of 0.2 pc, which would make the putative NSC in QSO1 more than 1 dex more concentrated than the

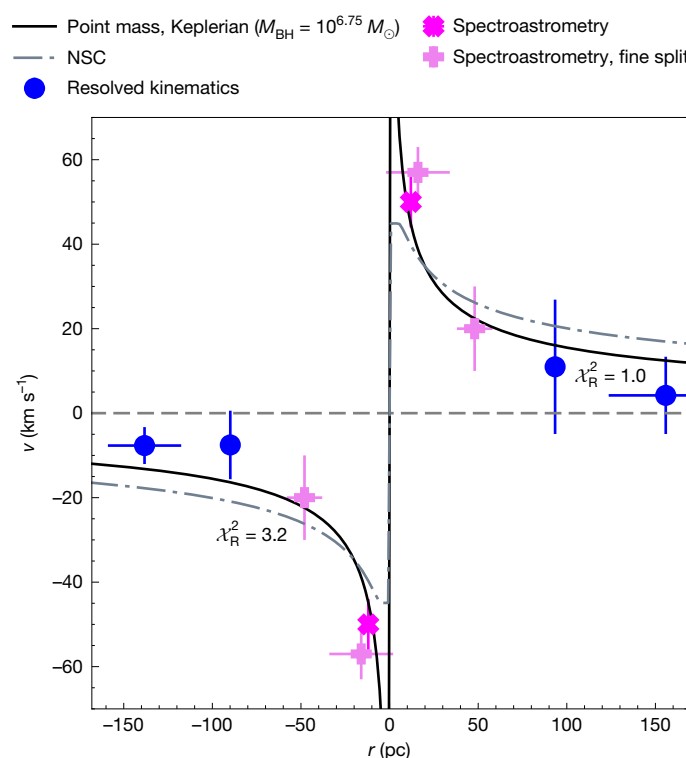

**Fig. 2 | One-dimensional rotation curve of QSO1.** Blue points are from the binned two-dimensional line-of-sight velocity field and the magenta crosses are the spectroastrometry measurements with lighter violet markers showing spectroastrometry of a finer split line profile, discussed in Methods. Line-of-sight velocity is given by $v$, while $r$ represents the distance from the dynamical centre. The solid black line indicates the Keplerian best fit with a point mass, giving a BH mass of $5.6 \times 10^6 M_\odot$ (which is a lower limit given that with this method the inclination is not constrained). The dot-dashed line is for an NSC (see text), which results in a worse fit.

densest nuclear clusters found in the local Universe, and also more concentrated than the densest star clusters found in the early universe (Extended Data Fig. 2).

In addition to the NSC, in Methods we also test the Plummer sphere model[20], which typically describes globular clusters, as well as a nuclear dark-matter cusp, both of which run into the same problems as the NSC—fitting the data by collapsing to a point mass.

Therefore, the simplest and most direct interpretation of the rotation curve is a central point mass larger than $\log(M_{BH}/M_\odot) = 6.75 \pm 0.15$, consistent with the bare spectroastrometric estimate, and corresponding to a supermassive BH. Once again, as the inclination of the rotation is unconstrained by this method, this mass estimate is a lower limit.

To examine the robustness of our conclusions against sources of systematic uncertainty, and without having to rely on spectroastrometry, we re-analyse our data by constructing self-consistent kinematics models by using the MOKA3D framework[21,22], which takes into account the detailed flux distribution of the kinematic tracers, inclination effects and smearing by the point spread function. The mass distribution models considered are the same as in the direct fitting above—the point mass, NSC and Plummer sphere. The model and residuals for the point-mass case are shown in Fig. 1, and the residuals in the cases of the NSC and the Plummer sphere are shown in Extended Data Fig. 3. Through this independent analysis, we find that the best fitting model ($\chi_R^2 = 1.17$) is Keplerian rotation around a point mass of $\log M = 7.7 \pm 0.3$, when corrected for $52 \pm 2°$ inclination estimated via the same method

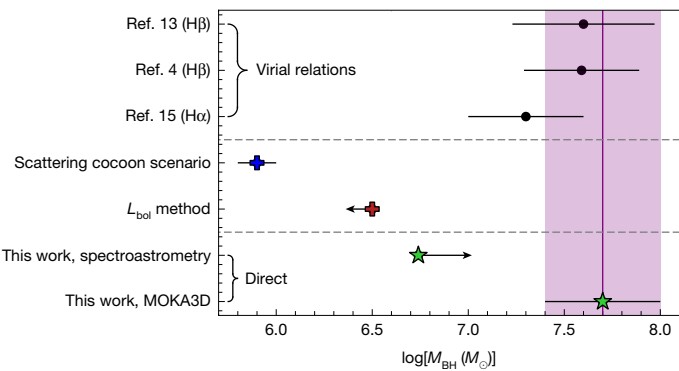

**Fig. 3 | Summary of BH mass estimates for QSO1.** Comparison between our lower limit and MOKA3D direct measurements (green stars) with previous virial estimates (black circles)[4,13,15], the scenarios assuming that the dominant broadening mechanism of the broad lines is due to electron scattering[7] and the $L_{bol}$ based estimates from ref. 36. The purple shaded region shows the $1\sigma$ uncertainty on the MOKA3D estimate.

(Methods), which is consistent with the lower limit obtained above. The Plummer sphere model can also provide a good fit ($\chi_R^2 = 1.60$); however, it does so by collapsing the sphere to a point mass with a best-fit scale radius $R_p = 0^{+3}_{-0}$ pc (Methods), just as for the direct one-dimensional rotation fitting discussed above. The NSC model does remain extended ($R_c = 4 \pm 2$ pc) when fitted to the full two-dimensional kinematics; however, the considerable systematic residuals (Extended Data Fig. 3, bottom) and the much higher $\chi_R^2 = 2.26$, indicate that this model is an inadequate fit to the data. Finally, we also robustly exclude kinematic contamination from outflows utilizing a combination of spectroastrometry and MOKA3D modelling (Supplementary Information section 1.5).

These results represent a direct, dynamical measurement of a BH mass at $z > 5$ and in an LRD. An immediate implication is that alternative scenarios explaining this LRD without accreting BHs are essentially ruled out. In addition, we can investigate the reliability of single-epoch BH mass virial estimates out to the epoch of reionization, and specifically for LRDs. Figure 3 shows a comparison between our BH mass direct measurement and literature estimates obtained using the virial scaling relations, showing full consistency. However, the scenario in which the broad lines are produced by electron scattering[7], while providing a good spectral fit to the broad lines (Supplementary Fig. 8), underestimates the BH mass by nearly 2 dex (Fig. 3).

The resulting Eddington luminosity ($L_{Edd}$) of the BH is $7.6 \times 10^{45}$ erg s$^{-1}$. By using standard scaling relations between broad H$\alpha$ and bolometric luminosity ($L$)[23], we infer that the BH is accreting well below its Eddington limit, with $L/L_{Edd} \approx 0.02$. If the broad H$\alpha$ to bolometric luminosity relation is higher than estimated locally (for example, ref. 6), then this would imply $L/L_{Edd} \approx 0.01$ or lower, indicating that the BH might be in a near-dormant state. For this reason, the BH mass estimates using bolometric luminosity ($L_{bol}$) with assumed $L/L_{Edd} = 1$, fail to match our dynamical measurement (Fig. 3). Yet, the BHs may still have experienced previous super-Eddington bursts, as inferred by ref. 24 for another overmassive, dormant BH at a similar redshift.

Finally, we note that the Keplerian rotation curve leaves little room for any stellar component. Specifically, in Supplementary Information section 1.7, we conservatively derive a dynamical upper limit on the stellar mass in the host galaxy of $M_* < 2 \times 10^7 M_\odot$. To our knowledge, this upper limit makes QSO1 the most 'naked' massive BH ever found, with $M_{BH}/M_* > 2$, and in line with the previous finding that this BH is hosted in an environment that is chemically nearly pristine[11]. This demonstrates the possibility of BH primacy, that is, BHs forming and growing

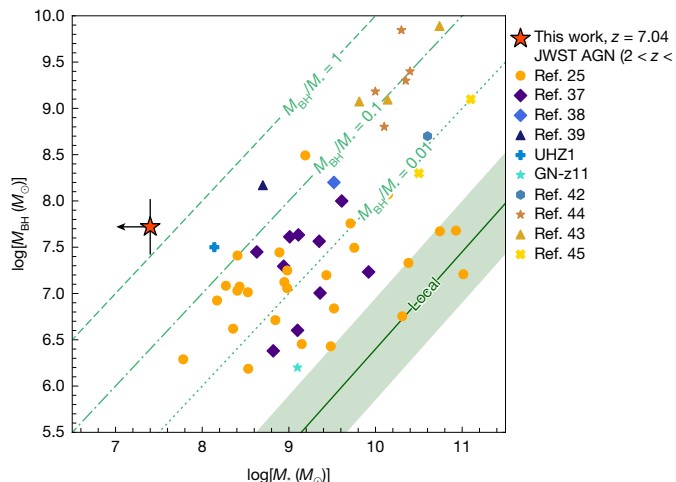

**Fig. 4 | Location of QSO1 on the $M_{BH}$–$M_*$ plane.** QSO1 is indicated by the red star. The remaining points represent measurements from other JWST observations of low-mass active galactic nuclei (AGN)[25,37–42] and quasars[43–45]. The solid green line shows the local scaling relation from ref. 17, with the scatter indicated by shading. The other green lines indicate constant $M_{BH}/M_*$ ratios. With $M_{BH}/M_* > 2$, QSO1 lies orders of magnitude above the local scaling relations and is approximately 1 dex more overmassive than even the most extreme sources found by JWST so far.

earlier than their host galaxy. The lower limit on the $M_{BH}/M_*$ ratio is three orders of magnitude higher than observed locally. Figure 4 shows how extreme QSO1 is on the $M_{BH}$ versus $M_*$ diagram relative the local relation—located at the extreme tail of the overmassive BHs identified by JWST in previous surveys[25].

The only scenarios that can account for such a system are those invoking 'heavy seeds', such as direct-collapse BHs (resulting from the direct collapse of massive pristine clouds) or primordial BHs (formed in the first second after the Big Bang)[26–28].

Yet, most direct-collapse BH scenarios would require a strong source of ultraviolet radiation in the vicinity, which is not seen (not even a post-starburst galaxy that might have produced ultraviolet radiation in the past), although some scenarios may expect direct collapse in different environments[29]. However, direct-collapse BH models suggest that their early growth is limited by the baryon fraction in an atomically cooling halo[30,31], placing an upper limit on the $M_{BH}/M_{dyn}$ ratio of approximately 0.1, that is, more than 1 dex lower than our inferred lower limit.

However, some independent evidence for the primordial BH scenario comes from the very low metallicity of this system[11]. However, the observed mass of $5 \times 10^7\ M_\odot$ is significantly higher than the preferred $10^6\ M_\odot$ primordial BH mass scale given by the electron–positron annihilation epoch in the ultra-early universe[32] (but see also ref. 33); therefore, the observed mass would require either significant accretion or rapid merging of primordial BHs, which may be linked to their highly clustered nature[34].

Regardless of the specific model, the high mass in such a remote cosmic epoch, the extremely high $M_{BH}/M_*$, together with the near-pristine environment[11], indicate that QSO1 is a massive BH seed caught in the earliest phases of accretion[35].

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

## Methods

### Conventions

Throughout this work we assume a flat $\Lambda$ (dark energy) cold-dark-matter cosmology with matter density parameter $\Omega_m = 0.315$ and a Hubble constant $H_0 = 67.4$ km s$^{-1}$ Mpc$^{-1}$ (ref. 46). All reported magnitudes are in the AB system. Following the lensing model of ref. 13, we adopt a flux magnification factor $\mu = 6.2$ and a shear factor of 3.52 for our source (image A of QSO1). Hence, 1 arcsec in the image plane corresponds to 1.52 physical kiloparsecs. For robustness tests, we use the Bayesian information criterion (BIC), defined as BIC $\equiv \chi^2 + k \ln n$, where $k$ is the total number of model parameters and $n$ is the number of points fitted; a decrease in BIC, $\Delta$BIC $\geq 5$, between two models was required for robust preference of one over the other, although our main conclusions remain unchanged even if a stricter $\Delta$BIC $\geq 10$ threshold is adopted.

### Data reduction

We use data from the BlackTHUNDER NIRSpec integral field unit survey, focusing on the 7.3-hour exposures with the G395H grating, giving a nominal spectral resolution $R = 3,700$ at the wavelength of H$\alpha$[15]. The NIRSpec integral field unit was centred on image A of QSO1 (right ascension 00:14:19.161; declination −30:24:05.664)[12]. A detailed description of the reduction procedures is available in refs. 4,15; however, a summary is provided here for context.

The spectra were extracted following the procedures of ref. 47, but using version 1.17.1 of the JWST pipeline. At $z = 7.04$, the H$\alpha$ line falls just outside the nominal wavelength coverage; however, the F290LP filter does not cut off longer wavelengths and the detector efficiency allows to recover H$\alpha$ emission. We perform this recovery by extrapolating the wavelength solution, flat-field curves and the grating-equation-derived line spread function (LSF) out of the nominal range and towards the detector sensitivity limit of $\lambda = 5.34$ μm. The peak of the H$\alpha$ line of QSO1 falls on $\lambda = 5.278$ μm; hence, our modification readily recovers the entirety of H$\alpha$ emission. Although flux calibrations beyond the nominal range may suffer inaccuracies, the primary interest of this work is a kinematics study; hence, our key kinematics results are insensitive to flux calibrations. The BH mass measurements are more affected. However, the square-root dependence of the BH mass on luminosity means that flux calibrations have to be wrong by an order of magnitude to significantly impact the measurements.

The nominal spaxel scale of the processed data was 0.05″; however, utilizing the large number of dithers, we are able to oversample the cube to a scale of 0.02″ per spaxel without incurring significant sampling artefacts. We choose the 0.02″ cube for the main kinematic and spectroastrometric analysis, with the 0.05″ cube used to perform consistency checks, ensuring that our results are not pixel sampling artefacts.

### Spectroastrometry of the rotation curve

To constrain the density profile of QSO1, we combine spectroastrometric measurements with resolved kinematics. The technical details of spaxel-by-spaxel fitting and spectroastrometry are given in Supplementary Information sections 1.1 and 1.2; here we summarize that we subtract the broad H$\alpha$ emission from the cube and create images of different velocity channels of narrow H$\alpha$ for which centroids can be obtained at sub-point-spread-function scales (provided a sufficient signal to noise[48]) and used to map dynamics below the nominal instrumental resolution[49,50]. The fiducial spectroastrometric analysis utilizing two velocity channels for higher signal to noise is shown in Extended Data Fig. 1. However, as shown in Fig. 2, splitting the line into finer bins does not change our results.

We infer the outer parts of the rotation curve by binning the line-of-sight velocity field (shown in Fig. 1) on scales >60 pc to avoid beam smearing. This procedure resulted in 4 bins covering the negative and positive sides of the rotation curve with $\langle v \sin i \rangle \approx 10$ km s$^{-1}$ (where $v$ is the line-of-sight velocity and $i$ is the inclination angle) with nominal uncertainties of order 1 km s$^{-1}$. However, these uncertainties, estimated

through the standard root mean square (rms) weighting scheme, do not take into account the velocity field cross-correlation between spaxels of each bin owing to beam smearing and hence are probably significantly underestimated. An a priori derivation of the covariance matrix is intractable as it would require fitting individual dithers, which have far too low signal to noise. We thus use an empirical approach—scaling the naive rms-derived errors until the optimal model in the family of models fitted has $\chi_R^2 = 1$. This yields an upper limit on the possible errors as it assumes that the optimal model is the ground truth. Consequently, using this method, we can establish a lower limit on the significance of the optimal model (which turns out to be a point mass) over other models considered.

For the spectroastrometric data points, we use a flux weighted average of the velocity channels, giving $\langle v \sin i \rangle = 51 \pm 4$ km s$^{-1}$. The factor of $\sin i$ is written to explicitly state that these are the projected values, uncorrected for inclination.

As the resultant rotation curve, shown in Fig. 2, is sparsely sampled, we consider only simple one- or two-parameter models for fitting. Model curves were constructed following a Keplerian prescription:

$$v(R) = \sqrt{\frac{GM(<R)}{R}} \qquad (1)$$

where $G$ is the gravitational constant, $R$ is the distance from the centre and $M(<R)$ is the mass enclosed within $R$. Our fiducial fit follows a point-mass assumption with $M(<R) \equiv M \equiv$ constant, which yields $\log(M/M_\odot) = 6.75 \pm 0.05$. We note that the uncertainty on this value is purely a fitting error and could be underestimated; hence, we re-estimate the error using bootstrap resampling, taking into account the width of the velocity bins, and we estimate a more conservative measurement error of 0.15 dex.

To fit the curve with a compact stellar-mass distribution, we fit the data with an NSC model derived for the Milky Way by ref. 19 who find a density profile following an $R^{-2}$ power law in the central 5 pc and dropping off as $R^{-3}$. From this density profile, we construct the following function for $M(<R)$:

$$M(<R) = \begin{cases} 4\pi RA & \text{if } R < R_c \\ 4\pi R_c A \left[ 1 + \log\left(\frac{R}{R_c}\right) \right] & \text{if } R \geq R_c \end{cases} \qquad (2)$$

where $A$ is the parameter setting the overall normalization and $R_c$ is the radius at which the switch in the power-law profile occurs. It is important to note that this is different from the 'effective radius' $R_e$ of the two-dimensional light distribution; in the case of the Milky Way NSC, the $R_e$ is a factor of 0.84 smaller than $R_c$. We initially fit the NSC model fixing $R_c$ to 5 pc, the same value as found by ref. 19. This fit is shown as a dashed grey line in Fig. 2 and produces a considerably worse $\chi_R^2 = 3.2$ than the point-mass (pure BH) fit with $\chi_R^2 = 1.0$. This corresponds to a difference in the BIC of about 11, indicating robust preference for the point-mass fit, particularly as our error rescaling procedure provides a lower limit on the significance of the preferred model. If $R_c$ is allowed to vary freely, then the NSC best-fit model gives $R_c = 10^{-4}$ pc with $M(<R_c) \approx 10^6 M_\odot$, which would imply extreme stellar densities, in excess of $10^{17} M_\odot$ pc$^{-3}$. Such densities are orders of magnitude above the densest stellar systems seen in the Universe and show that our NSC model effectively collapses to a point mass if $R_c$ is not fixed. We estimate an upper limit on $R_c$ by fitting a fixed value and lowering it until the difference in BIC between the best-fit and the fixed $R_c$ model reduces below 5. This way we estimate $R_c < 0.2$ pc with $M(<R_c) \approx 10^{6.2} M_\odot$; this limit is over 1 dex below even the most compact NSC in this mass range in the local Universe[51], as well as the dense star clusters found in the lensed Cosmic Gems arc by ref. 52, as illustrated in Extended Data Fig. 2. The upper limit on the NSC stellar mass is even lower if one adopts the density profile inferred for NSCs in other galaxies ($\rho \propto r^{-2}$)[53]. Hence, a point mass is needed to account for our observed dynamics.

In addition to the NSC model described above we consider the Plummer sphere[20] model, frequently used to describe the density profiles of globular clusters. The enclosed mass function for the Plummer sphere takes the following form:

$$M(<R) = M_0 \frac{R^3}{(R^2 + R_P{}^2)^{3/2}} \qquad (3)$$

where $M_0$ is the total mass of the system and $R_P$ is the scale radius. Although this model performs similarly well to the point mass, it does so by fitting $R_P \approx 10^{-4}$ pc and thus results in similar unphysically high stellar densities, as in the case of the NSC.

In addition to being excluded by the $\Delta$BIC value, a Milky Way NSC-like density profile with $R_c \approx 5$ pc produces a total stellar mass of $10^{7.2}\,M_\odot$, which is above constraints on the stellar mass derived from ultraviolet and optical emission as discussed in Supplementary Information section 1.8.

A remaining potential caveat of our analysis is that only isotropic velocity distributions were considered—velocity anisotropies could steepen the radial velocity gradient of a diffuse mass component[54], increasing the allowable extended mass. However, the steepness of the observed velocity gradient is such that any extended component collapses to a point when the scale radius is left free. Hence, it is unlikely that anisotropies of the underlying velocity field significantly skew our results.

Lastly, we explore what, if any, constraints on the dark-matter halo surrounding the object can be obtained from our data. We thus fit the widely adopted Navarro–Frenk–White density profile[55]. The enclosed mass for which is given by:

$$M(<R) = 4\pi\rho_0 R_s^3 \left[ \ln\left(\frac{R + R_s}{R_s}\right) - \frac{R}{R + R_s} \right], \qquad (4)$$

where $\rho_0$ and $R_s$ are the characteristic density and scale radius, respectively. However, as with the previous extended mass distributions, the above model collapses to a point with $R_s \approx 10^{-4}$ and $\rho_0 \approx 10^{14}\,M_\odot\,\mathrm{pc}^{-3}$, once more collapsing to a point mass and producing unrealistic densities. However, this does not imply that QSO1 resides outside of a dark-matter halo. Instead, our attempts at reproducing the kinematics with an extended density profile imply that any extended mass component is sub-dominant at the <200 pc scales probed by our measurements. We attempt the above analysis using H$\beta$ narrow or [O III]. However, these lines are too faint for constraining measurements.

## MOKA3D kinematics modelling

The measurements described above, although self-consistent, do not fully take into account instrumental effects such as point-spread-function beam smearing and the emissivity distribution of the light tracers. In the simplified analysis above, we overcome these issues by leveraging spectroastrometry. However, to check the robustness of our conclusions if such effects are accounted for, we independently refit the narrow-line cube with the MOKA3D framework[21,22]. MOKA3D is a 3D kinematic framework that can model conical outflows or disks by assuming spherical, conical or cylindrical geometries respectively, with any irregular distribution of the emitting clouds within the velocity field. The 3D model is populated with a distribution of fictitious clouds that account for the observed emission. These clouds are weighted according to the observed narrow H$\alpha$ flux in each spaxel and spectral channel of the data cube. As described in the following, the model clouds follow an analytical velocity field as a function of the radius, whose parameters are fitted to reproduce the observed emission and kinematic features via least-squares minimization.

As in the previous analysis, we consider three potential mass distributions—a point mass, an NSC density profile and a Plummer sphere. For each potential mass distribution, we parameterized the model clouds' circular velocity following equations (1)–(3). The free parameters of each kinematic profile are listed in Extended Data Table 1, specifically: total mass (in the case of the Keplerian rotation, this is the point mass);

parameter $A$ from equation (2), which represents the overall normalization of the NSC model; inclination of the disk (0° would be a face-on disk) or inclination of the outflow axis (0° outflow pointing towards the observer); effective radius of the density distribution (defined by equations (3) and (2) for the Plummer sphere and NSC models, respectively); intrinsic velocity for the outflow model; and the position angle of the kinematic configuration on the plane of the sky (measured clockwise from the top of the image). As discussed in ref. 21, to remove the degeneracies that are present when deriving 3D structures from the observed two-dimensional projections on sky, we minimized the numbers of free parameters by considering pure circular motions with no radial flows (except when testing the outflow scenario). For each potential mass distribution, we allowed the free parameters to vary in a wide range with no a priori constraints. Then, we ran the fitting routine—creating a 3D model cube with the same spatial and spectral binning as the observed data for each set of parameters. We then extracted the integrated model spectrum and compared with the observed one. The optimal set of parameters for each model was derived following the least-squares method of minimizing the distance between the observed and the model data cubes. During this procedure, we convolve the model cube with the point spread function and weight each fictitious cloud in the 3D model cube by the observed flux in each spectral channel of each pixel to ensure that any mismatch between model and data is solely due to assumptions on the underlying geometry and kinematics. This procedure allows us to obtain a 3D model cube—identical to the observed one in terms of spatial and spectral binning—from which we can compute moment maps to be compared with observations. In-depth descriptions of the principles being MOKA3D are provided in refs. 21,22.

The comparison between the residuals of the different models is given in Fig. 1 (for the point-mass case) and Extended Data Fig. 3 (for the NSC and for the Plummer sphere cases). As can be seen in the figures, a point-mass profile is strongly preferred over any extended density profile. The Plummer sphere model, while leaving similar residuals to a Keplerian curve, does so with a scale radius $R_P = 0^{+3}_{-0}$ pc (Extended Data Fig. 3), essentially collapsing into a BH. The NSC model remains extended even when $R_c$ is allowed to vary (best-fit $R_c = 4 \pm 2$ pc). However, it leaves significant systematic residuals in both velocity and velocity dispersion profiles, as shown in Extended Data Fig. 3.

As discussed in the following, very little velocity dispersion is included in the modelling (less than 7 km s$^{-1}$, well below the instrumental resolution) as the velocity dispersion in each pixel can be very simply reproduced by correctly modelling the line profile in each spaxel and weighting the model clouds against the observed flux; in other words, the apparent velocity dispersion is mostly and fully recovered simply by the velocity field, beam smearing and brightness distribution. In particular, a precise determination of the best-fit parameters via MOKA3D, combined with the innovative flux-weighting technique exploited by this model, guarantee to reproduce the observed moment maps at unprecedented detail, as demonstrated by refs. 56–58. Indeed, by definition, moment maps are computed taking into account the flux distribution along the velocity space, which is what makes MOKA3D so effective in reproducing them once the observed and model line profile match. The crucial step of assigning the observed flux in each spectral and spatial channel to the corresponding model clouds that belong to the same channel is what allows MOKA3D to reproduce the observed feature with such high accuracy, once the model parameters are correctly inferred. It is important to stress that no flux distribution is ever assumed by MOKA3D but instead is recovered from the data via the weighting procedure and after the correct set of parameters is unveiled. This allows MOKA3D to reproduce extremely irregular and asymmetric features. Therefore, despite the set of best-fit parameters inferred with MOKA3D, any extended density profile is not the preferred parameterization to reproduce the observed features in each spaxel due to absence of model clouds with the necessary projected velocity.

To further clarify that the underlying MOKA3D kinematical model does not intrinsically have asymmetries, Extended Data Fig. 4 shows the intrinsic unweighted kinematical model underlying the full weighted model reported in Fig. 1. The asymmetric flux distribution (in the flux map) and asymmetric kinematic features (for example, in the velocity dispersion map) seen in Fig. 1 emerge only when MOKA3D weights each emitting clouds to optimally reproduce the observed (irregular) flux distribution.

It is notable that the MOKA3D point-mass model gives $\log(M_{BH}/M_\odot) = 7.7 \pm 0.3$ with an inclination of $52° \pm 2°$, entirely consistent with the lower limit obtained from the previous direct measurements. In fact, the spectroastrometric mass estimate, when corrected for the inclination becomes consistent with the MOKA3D value to within $2\sigma$ ($6.9$–$7.2$ versus $7.7 \pm 0.3$).

To test whether the inclination estimate is robust, as well as to verify the presence of a rotating disk, we construct a non-parametric model wherein the disk is split up into three distinct shells that are fitted with independent inclinations. We find that this model fits the data well and produces shell inclinations of approximately $45 \pm 10°$, consistent with each other and the value found by the parametric models (Extended Data Fig. 5). The fact that consistent results are obtained regardless of the precise analytical procedure used indicates that our measurements are robust. In Supplementary Information sections 1.4 and 1.5, we further consider, and rule out, contributions of outflows to the narrow-line kinematics.

We also note that a diffuse, extended (exponential) disk-like mass distribution ($M \approx 10^7\,M_\odot$, $R_s = 150$ pc), while reproducing the large-scale ($r > 100$ pc) kinematics, would predict a smoothly declining velocity towards the centre, in stark contrast to the steep inner velocity slope of the one-dimensional curve, as illustrated in Extended Data Fig. 6. Likewise, MOKA3D gives substantial residuals for such an extended density profile (Extended Data Fig. 7).

We finally emphasize that the MOKA3D analysis is fully independent of the spectroastrometry approach. Although both analyses use the same data cube as input, they are otherwise entirely separate approaches with spectroastrometry hinging on the simple approach of centroiding different velocity channels, whereas MOKA3D self-consistently models the entire data cube.

## Data availability

The data used in this study were obtained as part of JWST programme ID 5015, and are available from the Mikulski Archive for Space Telescopes (https://mast.stsci.edu/portal/Mashup/Clients/Mast/Portal.html) at the Space Telescope Science Institute, which is operated by the Association of Universities for Research in Astronomy, Inc., under NASA contract NAS 5-03127 for JWST. The exact reduced data products used are available on Zenodo (https://zenodo.org/records/19402518).

## Code availability

The MOKA3D code underpinning this study is available at https://github.com/cosimomarconcini/moka_3d. A simplified demo version capable of reproducing the main results is made available with the paper.

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

**Acknowledgements** We thank P. Natarajan for comments. I.J. acknowledges support by the Huo Family Foundation through a P.C. Ho PhD Studentship. This work is based on observations made with the National Aeronautics and Space Administration (NASA)/European Space Agency (ESA)/Canadian Space Agency (CSA) JWST. The data were obtained from the Mikulski Archive for Space Telescopes at the STScI, which is operated by the Association of Universities for Research in Astronomy, Inc., under NASA contract NAS 5-03127 for JWST. These observations are associated with programme PID 5015. R.M., F.D'E., J.S., I.J. and G.C.J. acknowledge support from the Science and Technology Facilities Council (STFC), by the European Research Council (ERC) through Advanced Grant 695671 'QUENCH', by the UK Research and Innovation (UKRI) Frontier Research grant RISEandFALL. R.M. also acknowledges support from a Royal Society Research Professorship grant. S.A. acknowledges grant PID2021-127718NB-I00 funded by the Spanish Ministry of Science and Innovation/State Agency of Research (MICIN/AEI/ 10.13039/501100011033). H.Ü. acknowledges funding by the European Union (ERC APEX, 101164796). Views and opinions expressed are however those of the authors only and do not necessarily reflect those of the European Union or the European Research Council Executive Agency. Neither the European Union nor the granting authority can be held responsible for them. A.J.B. acknowledges funding from the 'FirstGalaxies' Advanced Grant from the European Research Council (ERC) under the European Union's Horizon 2020 research and innovation programme (grant agreement number 789056). S. Carniani acknowledges support by European Union's HE ERC Starting Grant No. 101040227 - WINGS. S.K. has been supported by a Junior Research Fellowship from St Catharine's College, Cambridge and a Research Fellowship from the Royal Commission for the Exhibition of 1851. R.S. acknowledges support from the PRIN2022 MUR project 2022CB3PJ3 - First Light And Galaxy aSsembly (FLAGS) funded by the European Union - Next Generation EU and from the EU-Recovery Fund PNRR - National Centre for HPC, Big Data and Quantum Computing. V.B., B.L. and S.Z. acknowledge the Texas Advanced Computing Center (TACC) for providing HPC resources under allocation AST23026. D.S. acknowledges support from the STFC, grant code ST/W000997/1. A.T. acknowledges support from the PRIN MUR project 2022935STW, funded by European Union - Next Generation EU, and from the INAF Fundamental Research 2023 Mini-grant project 'Cosmic Archaeology with the first black hole seeds'. G.C. acknowledges support from the INAF GO grant 2024 'A JWST/MIRI MIRACLE: MidIR Activity of Circumnuclear Line Emission'. K.I. acknowledges support from the National Natural Science Foundation of China (12573015, 1251101148, 12233001), the Beijing Natural Science Foundation (IS25003), and the China Manned Space Program (CMS-CSST-2025-A09). P.G.P.-G. acknowledges support from grant PID2022-139567NB-I00 funded by Spanish Ministerio de Ciencia e Innovación MCIN/AEI/10.13039/501100011033, FEDER Una manera de hacer Europa. M.P. acknowledges support through the grants PID2021-127718NB-I00, PID2024-159902NA-I00 and RYC2023-044853-I, funded by the Spain Ministry of Science and Innovation/State Agency of Research MCIN/AEI/10.13039/501100011033 and El Fondo Social Europeo Plus FSE+. R.V. acknowledges support from PRIN MUR '2022935STW' funded by European Union - Next Generation EU, Missione 4 Componente 2 CUP C53D23000950006 and from the Bando Ricerca Fondamentale INAF 2023, Theory Grant 'Theoretical models for Black Holes Archaeology. J.W. acknowledges support from the Cosmic Dawn Center through the DAWN Fellowship. The Cosmic Dawn Center (DAWN) is funded by the Danish National Research Foundation under grant number 140. I.J. discloses support for the research of this work from the Huo Family Foundation. The other authors declare no relevant funding.

**Author contributions** I.J. led the data analysis and writing of the paper with C.M. performing independent verification of the results. Key contributions to the data analysis were provided by F.D'E., R.M., A.M., H.Ü., J.S. and X.J. Observations were prepared and data reduced by F.D'E., H.Ü. and M.P. Theoretical interpretation of the data and discussion of the seeding models was contributed by J.S.B., V.B., P.D., A.F., K.I., S.K., B.L., J.L., R.S., D.S., A.T., R.V., M.V. and S.Z. G.C.J. provided the standard star data for the null test. S.M. verified the star cluster mass profile models. S.A., A.J.B., S. Carniani, S. Charlot, G.C., E.E., Y.I., L.R.I., N.L., G.M., E.P., P.G.P.-G., B.R., S.T. and J.W. provided comments on the paper and assisted in checking the validity of the results.

**Competing interests** The authors declare no competing interests.

**Additional information**
**Correspondence and requests for materials** should be addressed to Ignas Juodžbalis.

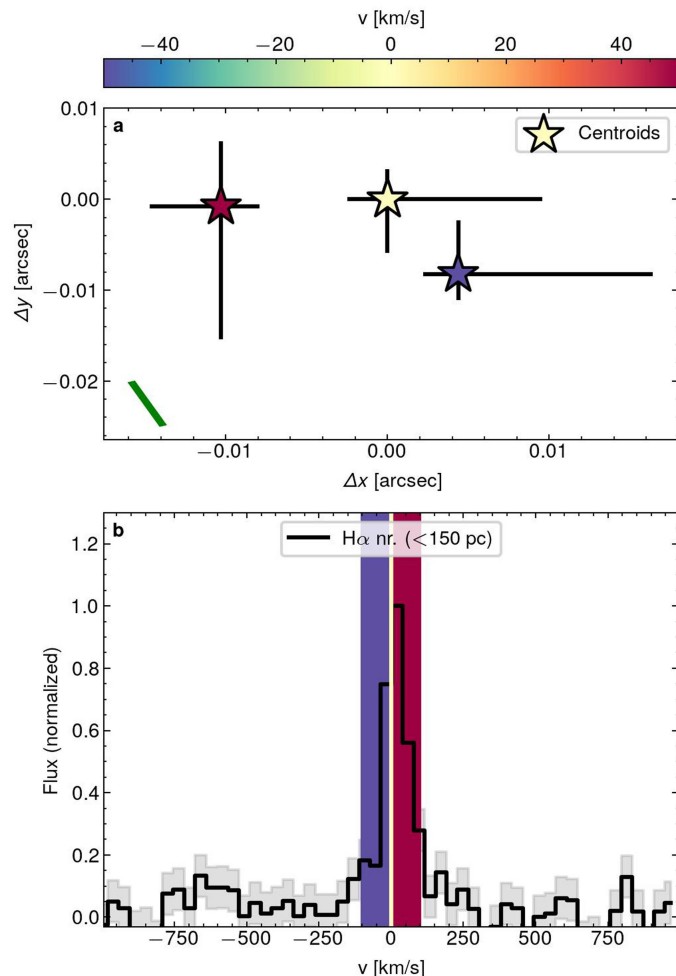

**Extended Data Fig. 1 | Spectroastrometry on the narrow line.** Panel **a** -
locations of the centroids of each velocity channel with the green line showing
the PA of the IFU slicer. Panel **b** narrow Hα line profile extracted from
a 0.1″ aperture (coinciding with the scales traced by our kinematics fits) with
colored bars showing the velocity bins. While the line profile appears to show
some evidence of a wing at ~ −150 km s⁻¹, expanding the velocity channels to
include it does not meaningfully alter the results.

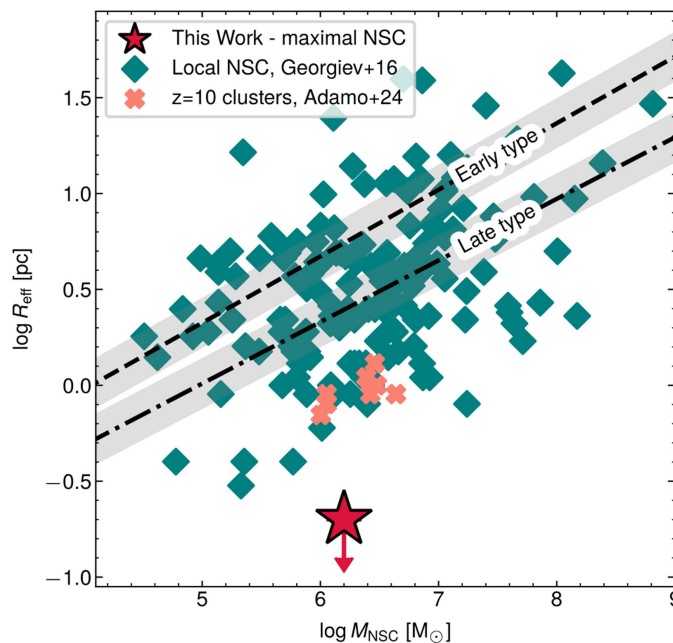

**Extended Data Fig. 2 | Comparison between QSO1 and star cluster observations on the mass-radius plane.** The upper limit on the size of a NSC in QSO1 is shown by a red star, illustrating that it is implausible when compared with local NSC (teal diamonds)[51] and dense star clusters at z = 10 (salmon crosses)[52]. The black lines illustrate mass-radius scaling relations for Early and Late type galaxies derived by ref. 51 with grey shading indicating scatter. The value for QSO1 represent the maximal extent allowed by an NSC-only model. The upper limit on the radius is nearly an order of magnitude lower than both local and distant clusters at a similar mass range.

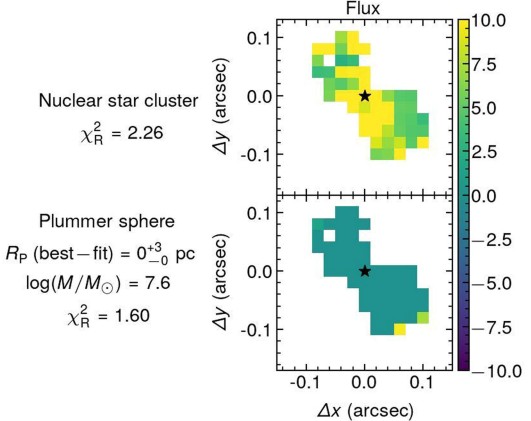
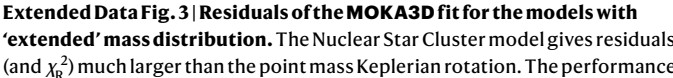
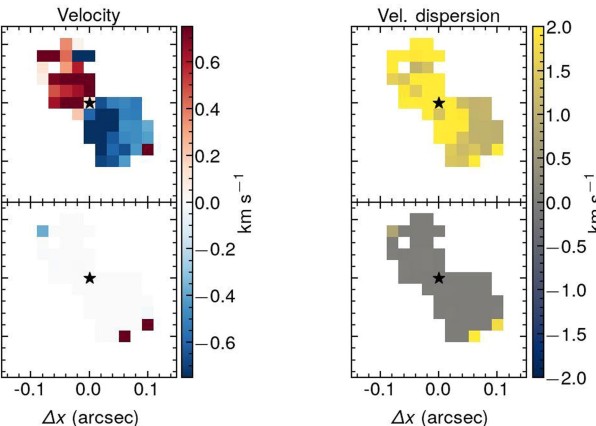

**Extended Data Fig. 3 | Residuals of the MOKA3D fit for the models with 'extended' mass distribution.** The Nuclear Star Cluster model gives residuals (and $\chi_R^2$) much larger than the point mass Keplerian rotation. The performance of the Plummer Sphere model is effectively only slightly worse than that of a point mass ($\chi_R^2 = 1.6$ vs $\chi_R^2 = 1.17$); however, the best fit $R_p = 0_{-0}^{+3}$ pc indicates that this is simply because the model ends up reproducing a point mass.

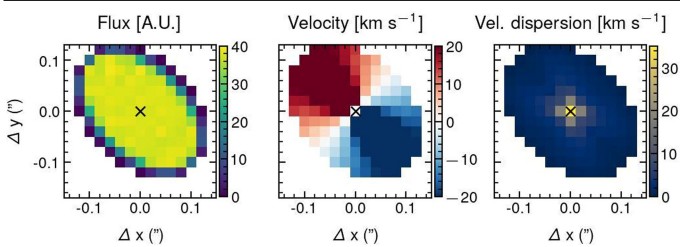

**Extended Data Fig. 4 | Intrinsic, unweighted, MOKA3D model.** From left to right - flux, velocity field and velocity dispersion. This model underlies the full weighted model shown in Fig. 1.

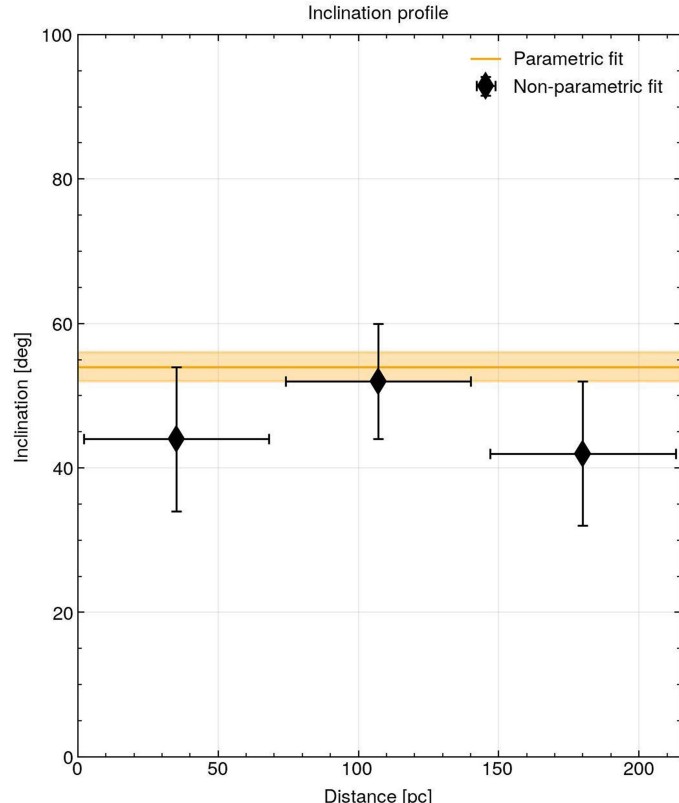

**Extended Data Fig. 5 | MOKA3D inclination constraints.** Inclination value found by fitting parametric models, orange line with shading indicating 1σ uncertainty, compared to inclinations found from a non parametric disk model in three independent rings (black points). The non parametric model is internally consistent among the three independent rings and consistent with the parametric curves well within 1σ.

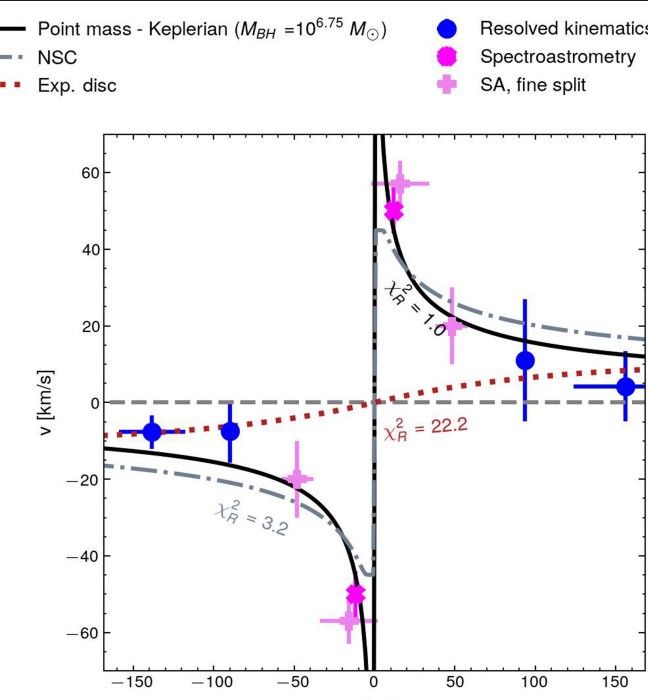

**Extended Data Fig. 6 | Spectroastrometric rotation curve with an exponential disc model.** Same as Fig. 2 in the main text, except the dotted line illustrates the velocity field expected from an exponential disc with ~$10^7$ M$_\odot$ and $R_s$ = 150 pc. As shown here, such a model provides a considerably worse fit than the others considered.

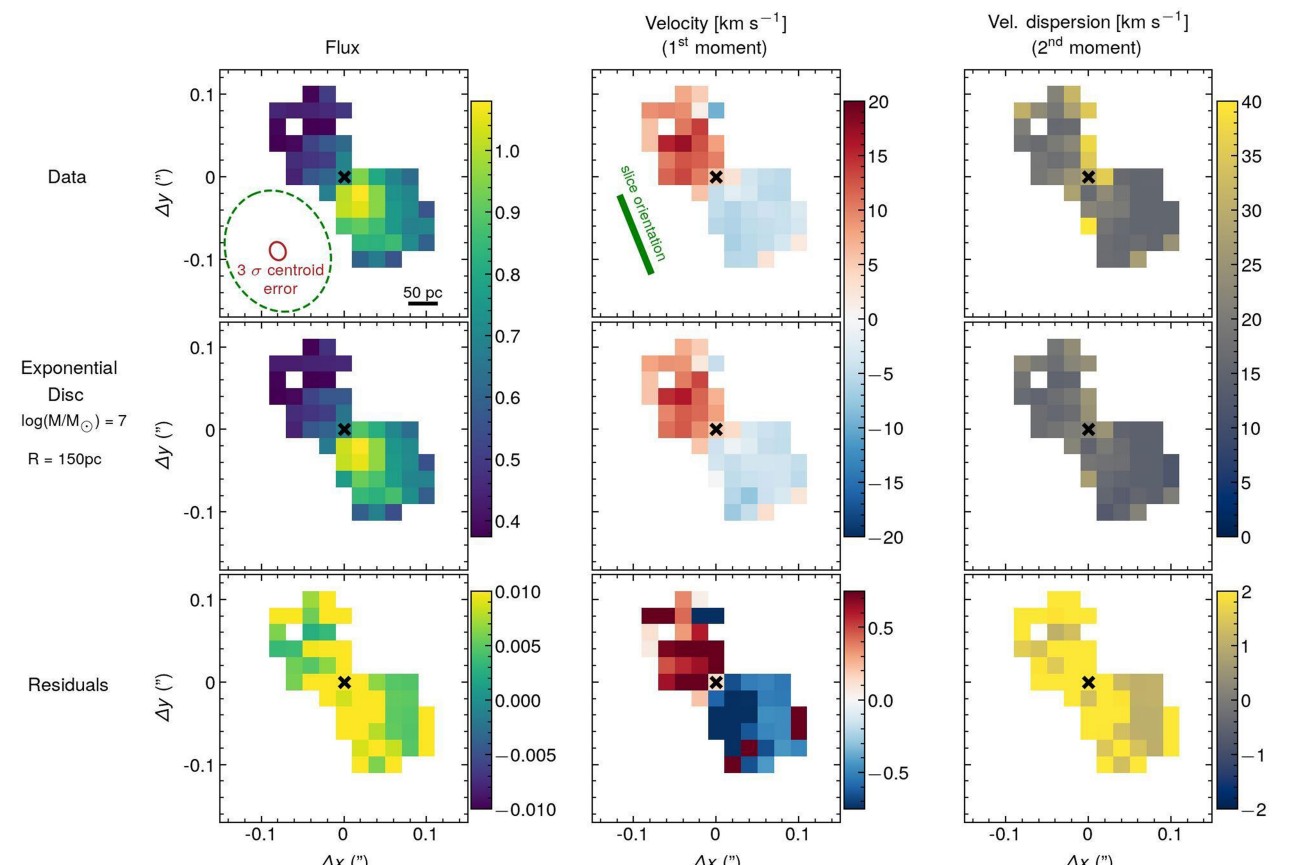

**Extended Data Fig. 7 | MOKA3D residuals for an exponential disc.** The disk was taken to have ~$10^7 M_\odot$ and $R_s = 150$ pc. It can be seen that such a model does not reproduce the data as well as the fiducial point mass model.

**Extended Data Table 1 | MOKA3D free parameters and their best-fit values for the kinematic models explored**

| Kinematic profile | $\log(M)$ [M$_\odot$] | $\log A$ [M$_\odot$/pc] | Inclination [°] | R [pc] | V$_{out}$ [km s$^{-1}$] | PA [°] |
|---|---|---|---|---|---|---|
| Keplerian | 7.7±0.3 | – | 52±2 | – | – | 48 ±3 |
| NSC | – | 5.4±0.3 | – | 4±2 | – | 48 ±3 |
| Plummer Sphere | 7.6±0.2 | – | – | $0^{+3}_{-0}$ | – | – |
| Outflow | – | – | 85±5 | – | 110±10 | 138±4 |

From left to right: logarithmic mass ($\log(M)$) of the mass distribution, normalization parameter of the NSC mass distribution ($\log A$, see Eq. (2)), inclination of the model with respect to the line of sight, scale radius (R), radial outflow velocity (V$_{out}$), and position angle (PA), of either the rotation disc's lines of nodes or of the outflow's axis, on the plane of the sky.