## [Peer Review File · Nature]

A direct black hole mass measurement in a Little Red Dot at high redshift

Corresponding Author: Mr Ignas Juodžbalis

Version 1:

Reviewer comments:

Referee #1

(Remarks to the Author)
Referee report

This manuscript presents a dynamical mass measurement of a central source in a Little Red Dot (LRD) at $z = 7.04$, based on spatially resolved gas kinematics. While the nature and mass of LRDs remain controversial, this study claims the detection of Keplerian rotation on 50–150 pc scales, reporting a 50-million-solar-mass black hole with negligible stellar contribution. The result is compelling, if confirmed, as it would place strong constraints on the nature of LRDs, the early growth of black holes, and the mass scale of black hole seeds.

The study appears to be the first attempt to determine the dynamical mass of a strongly lensed LRD using valuable JWST IFU data. While the manuscript is clearly written and persuasive, it is not evident whether the data quality is sufficient to measure gas kinematics at each spaxel, nor whether the authors' approach to determining the dynamical mass is fully valid. To support the claimed validity of the dynamical mass, detailed fitting results and error analyses at the spaxel level are needed. In the MOKA analysis, only the final fitting results are shown, making it impossible to judge whether the outcomes are acceptable. I believe a study of this importance should be presented as a full-length paper, with in-depth analysis and methodological details. Below I list several points to support my judgment.

1. Measurement uncertainty of velocity and velocity dispersion

A key result of this manuscript is presented in Fig. 2 with a Keplerian curve. From the spatially resolved kinematics, V is measured at 50, 100, and 150 pc scales by combining equal numbers of spaxels. For each spaxel, velocity and velocity dispersion are measured and added in quadrature. It is unclear how accurately velocity and velocity dispersion can be measured from each spaxel, which has $S/N \approx 5$. The authors argue that V can be measured well below the sampling of 20–25 km/s. However, velocity dispersion cannot be reliably measured with this S/N . According to the authors, the spectral resolution is $\text{FWHM} = 81 \text{ km/s}$ ($\sigma \approx 34 \text{ km/s}$). In comparison, the reported σ of $H\alpha$, particularly for the blue-shifted component (bottom-right in the IFU image), is mostly $<10 \text{ km/s}$. It seems likely that the final error of V in Fig. 2 is underestimated.

The authors should provide several examples of $H\alpha$ line fitting, including the errors of velocity and velocity dispersion, as well as the error of their quadrature sum for each spaxel, to allow readers to assess the robustness of the error analysis. Note that in each spaxel, the value of velocity dispersion is comparable to or larger than the velocity, implying that the velocity dispersion error may dominate. Also, the reported S/N presumably refers to the spectrum before subtracting the continuum and broad $H\alpha$ component; thus, the S/N of the narrow $H\alpha$ line should be even lower. The authors should present the combined V (i.e., after adding σ of $H\alpha$) in addition to the IFU maps of flux, velocity, and velocity dispersion in Fig. 1.

Given the uncertainties in velocity and velocity dispersion, the reported residuals for the nuclear star cluster ($<1 \text{ km/s}$ in velocity and $<2 \text{ km/s}$ in velocity dispersion) are not meaningful. This can be better assessed once detailed error analyses are provided.

2. Kinematical constraints.

Taking the V values in Fig. 2 at face value, the resolved kinematics alone show no strong radial trend—the blue points are almost consistent with a flat distribution. If the errors are underestimated, this interpretation would be even stronger. Only when combined with the spectroastrometric result at 10–15 pc scales (pink points), the radial trend appears Keplerian.

I find the approach of combining a single spectroastrometric point with resolved kinematics incomplete and somewhat misleading. As the authors note, spectroastrometry provides advantages beyond the nominal spatial resolution, but it can also be applied to larger scales. It is unclear why a full spectroastrometric analysis independent of the resolved kinematics could not be performed. Instead of using one velocity channel ($0 < V < 100$ km/s), the authors could have used several channels to constrain a radial gradient. For example, the velocity channel $0 < V < 20$ –25 km/s and $25 < V < 50$ should have sufficient signal since it corresponds to the line peak. Such an analysis would provide an independent constraint on gas kinematics and could be compared with the resolved kinematic results. The authors may argue that the S/N is insufficient for more detailed velocity channel analysis, but if that is the case, it raises doubt about the feasibility of the spatially resolved analysis itself.

3. Comparison with a Keplerian curve.

It is misleading to denote the combined velocity and velocity dispersion as V (e.g., in Fig. 2), since readers might confuse it with the line-of-sight velocity. It should instead be expressed differently, for example as $\sigma_{\text{total}} = (V^2 + \sigma^2)^{1/2}$.

It is unclear whether directly comparing σ_{total} to a Keplerian curve is appropriate. While the velocity dispersion should indeed be added to the rotational velocity to represent the total gravitational potential, σ_{total} would not follow a Keplerian profile if the pressure component dominates—which appears to be the case for this LRD.

4. Possibility of outflows

The IFU maps in Fig. 1 can naturally be interpreted as showing a biconical outflow, though the outflow velocity seems much lower than in typical local Seyfert galaxies with similar Eddington ratios. It is possible that a broad wing (outflow component) of the narrow line is not detected due to the low S/N.

The authors argue that outflows can be ruled out based on velocity channel analysis and MOKA fitting results, but the brief description provided is insufficient to assess this claim. The S/N in the high-velocity (>500 km/s) channels would be very low, as they correspond to the faint wings of the line profile. Given that outflows have wide opening angles and that the astrometric position is affected by emissivity distribution and extinction, Extended Fig. 5 does not convincingly constrain the scenario.

Regarding the MOKA fitting, the method allows flexible input parameters. The MOKA outflow fitting result appears significantly worse than the point-mass or NSC cases, but even the flux distribution in Extended Fig. 6 does not match well, raising the question of whether a better fit might be possible. I do not mean to imply that the authors performed poorly, but rather to emphasize that essential methodological details are missing from this manuscript.

Based on these assessments, I conclude that the authors should present a more detailed analysis in a full-length paper to convincingly demonstrate their results to the community.

Referee #2

(Remarks to the Author)

This is an important observational result, with deep consequences for understanding little red dots, as the authors discuss. There are only a couple of points I wanted some clarification on.

1) The spectroastrometric (SA) method measures the flux-weighted photocenter as a function of velocity channel, not the location of the gas producing a given velocity. A single point derived from SA is an effective, model-dependent (emissivity, geometry, inclination, etc) radius. Is the radius uncertainty for the SA point reflecting all of these issues?

2) What is the impact on the results if the SA point is not considered? If the rotation curve was measured firmly within the black hole sphere of influence, as the authors claim, then presumably the overall estimates would not hinge so strongly on the innermost point. It would be useful to repeat the fit without the SA constraint and report how the BH mass, inclination, and enclosed mass profile change. This will make the sensitivity on the SA point explicit rather than implicit.

3) The authors use σ_N as a proxy for stellar velocity dispersion to estimate the black hole sphere of influence as 270 pc. Notice that this is considerably larger than typical values (e.g., $\sigma_* = 150$ km/s gives ~ 8 pc for the quoted BH mass). What is the justification for using σ_N instead of σ_* ?

Referee #3

(Remarks to the Author)

The paper "A direct black hole mass measurement in a Little Red Dot at the Epoch of Reionization" reports the dynamical detection of a supermassive black hole of 10^7 suns in a triply imaged source at $z=7.04$. If the measurements are sound, then this would be a truly remarkable and very important result. To our knowledge at this moment, the only dynamical black hole mass measurements outside the Local Volume have been made in either very luminous quasars with GRAVITY or in a couple of lensed galaxies at Cosmic Noon, with billion solar mass black holes. This measurement would be even more remarkable as the nature of the Little Red Dots is still very much under debate. This paper infers a system that is completely black hole dominated, a truly remarkable result if true.

However, we feel that the paper should present both (a) more examination of the data, and possible systematic uncertainties in the emission line fitting, (b) a stronger set of tests on the fidelity of the spectro-astrometry, and finally (c) a much clearer exposition of the MOKA-3D fitting, including a full accounting of free parameters.

Major Point 1: Broad+Medium+Absorption+Narrow-line fitting. There are many components fitted to the H α line. Since the BH detection hinges on the velocity field of the narrow H α line, we would like to see:

- an investigation into the degeneracy between the fit to the absorber, the two broad components, and the resulting flux, velocity, and dispersion field of the narrow line. We appreciate that the broad components are unresolved, but apparently the absorber sits in front of the broad line (according to D'Eugenio+2025) and so we are not convinced that fixing the absorber in velocity and width across the cube is necessarily justified.
- an exploration of the outcome of allowing multiple absorbers, or a varying velocity and width.
- propagating forward the uncertainty from these fits into the resulting velocity measurements

Major Point 2: The spectro-astrometry seems like a brilliant way to improve the resolution in this case. However, we are not clear on how the error bars shown in Extended Data Fig. 1 are translated into the rotation curve presented in the main text. Please explore:

- the significance of the spatial offset in more detail. For instance, Abuter+2024 (GRAVITY) present a null test to prove that they have measured an actual spatial offset as a function of velocity. This would be a worthwhile type of test to perform here.
- the sensitivity of the spatial offset to the exact spectral division between red and blue?
- propagating the uncertainty in the absorption line fit into the uncertainty in the total narrow line profile

Major Point 3: From the information presented, it is very hard to understand both (a) what free parameters are involved in the MOKA-3D fit or (b) what about the velocity field causes such a strong preference for an unresolved point mass.

- Please plot the models as well as the residuals for the fits.
- Please provide a complete list of the free parameters constrained by MOKA-3D, and how well-constrained they are.
- Of specific concern, please describe whether the dispersion is a free parameter, and whether it is allowed to vary spatially. Typically in black hole mass determination papers based on gas velocity fields, there is significant attention paid to possible turbulent (non-gravitational) support.
- Please describe the physical model generating the flux distribution in each case. Given that the observed flux distribution is asymmetric, we do not fully understand why the rotating disk model does such a clean job with the flux distribution. Similarly, the dispersion field is not symmetric, which seems unusual for a rotating disk. Please explain how the MOKA-3D model fits the dispersion field so cleanly.
- It would be useful here to comment on the spatial resolution achieved in the maps fitted by MOKA-3D. It seems that the sphere of influence is not reached, so how are the constraints on M_{bh} so tight?

We find the text very clear and well-written, but do feel we need more clarity on the methods and uncertainties as outlined above to understand the robustness of the measurement.

Referee #4

(Remarks to the Author)

I co-reviewed this manuscript with one of the reviewers who provided the listed reports.

Version 2:

Reviewer comments:

Referee #1

(Remarks to the Author)

I thank the authors for carefully addressing my comments and suggestions in their revision of the manuscript and in the response letter. Although I still believe that a full analysis in a regular journal would be preferable given the complexity of the analysis and the limitations of the data, I recommend the manuscript for publication and leave the assessment of the results to the readers.

On Page 8 after Eq.1, there is a typo: "ar unresolved" should be "are unresolved".

Referee #2

(Remarks to the Author)

Regarding my third comment about using the narrow line velocity dispersion. For the sake of completeness please explicitly mention in the supplementary material that no stellar features are detected and given the that σ_N tracks the σ_{star} (quoting the appropriate reference), they end up with the estimated BH sphere of influence estimate.

Otherwise, I am happy with the author's response.

Referee #3

(Remarks to the Author)

We thank the authors for responding to our detailed comments. The additional information is much appreciated. However, in some cases the additional information has opened new questions for us. We have tried to summarize our outstanding major concerns below.

Point 1: Spectro-astrometry

The authors have suggested that their results hinge more on the MOKA-3D modeling than on the spectro-astrometry. If so, perhaps this comment can be disregarded, but we would appreciate hearing the thoughts of the authors on this new concern about the spectro-astrometry. We studied Stern, Hennawi, & Pott (2015), a detailed exploration of spectro-astrometry techniques for luminous quasars. They have a subsection 3.4.1 on contamination from the narrow-line region, that is gas that is more spatially extended than the broad-line region. In this subsection, they explain that if there is NLR emission on the spatial scale of r_{PSF} , then it will be very hard to filter out this emission. Because the position at a given frequency is r -weighted (see their Eq. 12), even a very small fraction of the photons emerging from such large scales will dominate the position measurement at that frequency.

Please address whether it is possible to measure sub-PSF positions using narrow-line gas, when there is demonstrably extended emission on the scale of the PSF.

Point 2: Velocity Field

If the MOKA-3D modeling is the critical measurement, then we find it even more important that the authors investigate realistic uncertainties in the narrow Ha velocity. We are convinced by the authors that it is fair to fix the absorber parameters from the spatially averaged fit. However, there is uncertainty in the velocity and width of both the absorber and the broad emission line fits that is covariant with the uncertainty in the narrow velocity field. Please take into account the uncertainty in the absorber and broad-line fits when fitting the narrow line (e.g., by sampling from the posterior for the absorber and broad-line fit).

Point 3: MOKA-3D Fitting

We thank the authors for adding more detail about the MOKA-3D fitting, but we still find ourselves a bit confused. The claim seems to be that a velocity gradient of ~ 10 km/s measured over a spatial scale of ~ 300 pc over two independent beams is sufficient to rule out a spatially extended distribution of mass. Is that in fact the claim? It seems impossible on the face of it. The inferred mass is $\sim 10^7$ Msun. 100-200 pc seems consistent with mass-size measurements for galaxies of this mass. This leads to our first question: could the authors rule out an extended (~ 150 pc-scale) disk with $M^* \sim 10^7$ Msun?

In terms of the presented fits, we are still confused about the following points:

(1) In the primary fit (Fig 1), why does the Keplerian point mass model have a flux asymmetry? Furthermore, can the authors explain how the Keplerian model reproduces the velocity dispersion map so precisely given the spatial resolution? In particular, there are a few disconnected pixels with $\sigma=40$ km/s on either side of the dynamical center. How does the model reproduce that pattern so closely after being beam-smearred? Perhaps it would help to show the unsmoothed (true) velocity and velocity dispersion distributions?

(2) A related question is why the flux distributions (which are not shown) and residuals for the other models do NOT match the observed asymmetry in flux. If a symmetric rotating Keplerian disk can have an asymmetric flux distribution, why not these other models?

(3) If the authors adopt more realistic velocity errors as suggested in Point 2, and fix the (arbitrary?) flux distributions to match what is observed, then are the spatially extended models presented here actually inconsistent with the data?

Referee #4

(Remarks to the Author)

I co-reviewed this manuscript with one of the reviewers who provided the listed reports.

Version 3:

Reviewer comments:

Referee #1

(Remarks to the Author)

No further comments.

== Following request to comment on concerns from, and response to, Reviewers 3+4 ==

I am inclined to support the authors. The technique of the spectro-astrometry should provide information below the spatial resolution limit.

The two velocity channel maps ($V \sim 50$ km/s & $V \sim -50$ km/s) should mainly contain the photons from the high velocity clouds at sub-psf scales since the clouds at 60-70 pc scales would have much lower velocity. In other words, while photometry images cannot separate the sub-psf scale photons, the velocity channel maps can preferentially select sub-PSF photons. The exact location of the ± 50 km/s channels will depend on the flux and velocity distribution of the NLR. However, it is reasonable to assume that the contamination from the low velocity clouds at the PSF scale would be insignificant in the high velocity channel maps. In my 1st report I asked authors to provide more velocity channels and they showed 'fine split' in Fig 2, which I believe demonstrates the concept.

Referee #3

(Remarks to the Author)

We thank the authors for considering our issues from the second review. However, we do not feel satisfied that our concerns were sufficiently addressed, and so we will try again here to be clearer.

Point 1 from the authors: "The referees are correct in pointing out that for a generic AGN host the flux from larger scales could drown out a signal from the BLR, which is what Stern, Hennawi & Pott were attempting to resolve. However, unlike Stern et al., in our study we are not attempting to spectroastrometrically map out the BLR, but rather focus on the NLR dynamics itself (that is why we subtract the broad line emission from the cube). Therefore, in our case all emission originates from similar spatial scales and we do not have the issue of the ~ 3 dex difference between the BLR and NLR radii, which is what drives the contamination in Stern, Hennawi & Pott (2015). Therefore, standard spectroastrometric considerations apply and it is possible to measure sub PSF scales as per Ivison (2007)."

Whether the gas is in what we call the broad-line region or the narrow-line region is immaterial. The issue at hand here is what spatial scales you believe you can resolve with spectroastrometry in this source. Stern's point is very simple. If there is gas on the scale of the PSF, then you will not be able to probe scales smaller than that.

The authors cannot have things both ways. Either you are trying to resolve structures below the PSF scale or you are not. Taking the numbers in the manuscript accounting for magnification, the PSF scale is roughly 60-70 pc. The authors claim that the narrow-line region is seen to be extended out to 200 pc scales from a prior paper. This measurement seems robust to us. The critical size measurement is claimed to be on a size scale of 10 pc. Therefore, this measurement is below the scale of the PSF AND the authors claim that there is H α emission arising from all scales between 10 pc and 200 pc.

Then, coming back to the Stern+ point, the issue is that the measurement will be heavily weighted towards the emission coming from the scale of the PSF. In other words, you cannot make a relative astrometric measurement on 10-20 pc scales, when you also have emission on 50-100 pc scales, as the authors claim that they do. If the authors cannot explain why the measurement is not heavily weighted to 60-70 pc scales as argued concisely in Stern+, then we now conclude that the spectro-astrometry should be removed from the paper as it is not robust/cannot be trusted.

We move now to Point 3: We thank the authors for the thin disk model plots, these are very helpful. We see indeed that the thin disk does a fine job at scales > 50 pc, and only breaks within the PSF scale. Thus, based on point 1, we do not believe the spectro-astrometry can rule out the disk model.

Coming now to the MOKA-3D fit, we see that the residuals for the simple thin rotating disk already have residuals at the 0.5 km/s level. This is well below the spectral resolution of the instrument, but also (we suspect) at a level that a small error in the absorption correction could easily account for. That brings us back to Point 2, the reliability of the velocity field as the basis for the MOKA-3D modeling.

While we again agree with the authors that the geometric argument for fixing the broad line shape and the absorber position/width/depth for all spaxels in order to isolate the narrow component is well motivated, our previous point was that this subtraction in its current form does not take into account the uncertainty in the velocity, dispersion, and depth of the absorption and its covariance with the broad line properties. Per Table 1 in D'Eugenio et al. 2025 (from which the best fits were adopted), there is ~ 10 km/s uncertainty in both the position and dispersion of the absorber. The authors are currently neglecting this uncertainty in the velocity field, and it is not clear that different absorber positions and dispersions will result in uniform shifts in velocity in different spaxels to produce the same velocity field as the best fit. We request that the authors account for this uncertainty by sampling from the posterior in the absorber/broad line fits and performing the narrow line fits

for each draw. We request a two-dimensional map showing the resulting uncertainty in the velocity and velocity dispersion resulting from incorporating honest uncertainty in the decomposition.

The authors must then demonstrate in the MOKA-3D framework that there is still a strong preference for an unresolved point mass given a more realistic accounting of the uncertainty in the velocity field. If the authors can demonstrate that there is a strong preference in χ^2 for the unresolved fit once these covariant uncertainties are accounted for, we will be convinced that the 2D fits can indeed be viewed as evidence for the measured black hole mass. Given that we are skeptical about the efficacy of spectroastrometry on these clearly resolved narrow lines, we believe that the entire claim now hinges on the robustness of the structure in the velocity field.

Referee #4

(Remarks to the Author)

I co-reviewed this manuscript with one of the reviewers who provided the listed reports.

We would like to thank the referees for providing extensive comments and suggesting tests which have helped improve the presentation and robustness of our results. Due to editorial concerns some of the extended analysis has been moved to Supplemental Material. However, we are open to switching parts between the Methods and the Supplemental Material if the referees find it appropriate.

Before addressing any of the individual points, we would like to clarify that MOKA3D and binned velocity field/spectroastrometry provide entirely *independent* measurements derived solely from the initial spectrum, which give consistent results. Hence, while the consistency of the two methods strongly supports the conclusions of the paper, if needed for sake of clarity, either of them could be left out without losing the robustness of the result.

A point by point response to the referee comments is available below.

Referee #1 (Remarks to the Author):

Referee report

This manuscript presents a dynamical mass measurement of a central source in a Little Red Dot (LRD) at $z = 7.04$, based on spatially resolved gas kinematics. While the nature and mass of LRDs remain controversial, this study claims the detection of Keplerian rotation on 50–150 pc scales, reporting a 50-million-solar-mass black hole with negligible stellar contribution. The result is compelling, if confirmed, as it would place strong constraints on the nature of LRDs, the early growth of black holes, and the mass scale of black hole seeds.

The study appears to be the first attempt to determine the dynamical mass of a strongly lensed LRD using valuable JWST IFU data. While the manuscript is clearly written and persuasive, it is not evident whether the data quality is sufficient to measure gas kinematics at each spaxel, nor whether the authors' approach to determining the dynamical mass is fully valid. To support the claimed validity of the dynamical mass, detailed fitting results and error analyses at the spaxel level are needed. In the MOKA analysis, only the final fitting results are shown, making it impossible to judge whether the outcomes are acceptable. I believe a study of this importance should be presented as a full-length paper, with in-depth analysis and methodological details. Below I list several points to support my judgment.

We appreciate the concerns of the referee about the need for more details on the data analysis in support of our findings. In the revised version we have leveraged the Methods and the Supplementary Material parts to provide a more extensive

description of the analysis, together with the various additional tests that were required. While a regular journal may potentially give some more room to expand on these aspects, we believe that the importance of our results deserves publication in a high impact, multi-disciplinary journal such as Nature.

1. Measurement uncertainty of velocity and velocity dispersion

A key result of this manuscript is presented in Fig. 2 with a Keplerian curve. From the spatially resolved kinematics, V is measured at 50, 100, and 150 pc scales by combining equal numbers of spaxels. For each spaxel, velocity and velocity dispersion are measured and added in quadrature. It is unclear how accurately velocity and velocity dispersion can be measured from each spaxel, which has $S/N \approx 5$. The authors argue that V can be measured well below the sampling of 20–25 km/s. However, velocity dispersion cannot be reliably measured with this S/N . According to the authors, the spectral resolution is $FWHM = 81$ km/s ($\sigma \approx 34$ km/s). In comparison, the reported σ of $H\alpha$, particularly for the blue-shifted component (bottom-right in the IFU image), is mostly <10 km/s. It seems likely that the final error of V in Fig. 2 is underestimated.

The authors should provide several examples of $H\alpha$ line fitting, including the errors of velocity and velocity dispersion, as well as the error of their quadrature sum for each spaxel, to allow readers to assess the robustness of the error analysis. Note that in each spaxel, the value of velocity dispersion is comparable to or larger than the velocity, implying that the velocity dispersion error may dominate. Also, the reported S/N presumably refers to the spectrum before subtracting the continuum and broad $H\alpha$ component; thus, the S/N of the narrow $H\alpha$ line should be even lower. The authors should present the combined V (i.e., after adding σ of $H\alpha$) in addition to the IFU maps of flux, velocity, and velocity dispersion in Fig. 1.

Given the uncertainties in velocity and velocity dispersion, the reported residuals for the nuclear star cluster (<1 km/s in velocity and <2 km/s in velocity dispersion) are not meaningful. This can be better assessed once detailed error analyses are provided.

We thank the referee for the suggestions. We would like to note that the reported S/N refers to the S/N of the fitted narrow $H\alpha$ component, i.e. *after* broad component (and continuum) removal, hence represents the S/N of the subtracted cube on which the analysis was performed - we have clarified this in the text and the caption of Fig. 1.

We also note that our fitting procedure, convolving the intrinsic line dispersion with instrumental resolution, is capable of robustly constraining whether the line is resolved

or unresolved - in the latter case the posterior ‘piles up’ in an estimate of the velocity dispersion that is consistent with zero.

To illustrate whether the line is resolved or unresolved, we construct significance maps of $\sigma/\Delta\sigma_-$, where $\Delta\sigma_-$ is the error on the dispersion on the low side. We consider the line resolved if this ratio is >5 and unresolved otherwise.

As shown in the figure, H α intrinsic width is recovered to good significance near the center of the would-be axis of rotation, while the line is completely unresolved in the outer parts, despite the S/N being slightly higher on the blue side.

Within this context, and regarding the comment that the “velocity dispersion is comparable to or larger than the velocity”, we note that in the central region, which is the most crucial for constraining the BH mass, the velocity dispersion is ~ 35 km/s while the velocity is $V_{rot} \cdot \sin i \sim 55$ km/s (i.e. about 70 km/s deprojected), so the velocity dispersion is lower than the velocity; in the outer region, beyond >120 pc, our velocity dispersion measurements are all upper limits, $\sigma < 10-20$ km/s, and anyway these outer parts are less critical in constraining the BH mass. Therefore, the dispersion does *not* dominate throughout the disk.

Even more importantly, as clarified in the revised version of the Methods, the MOKA3D analysis reveals that the observed velocity dispersion distribution is fully recovered by the simple velocity field convolved with the brightness distribution via

the beam smearing; in other words, MOKA3D illustrates that there is no need to add any intrinsic velocity dispersion to the model (in addition to the apparent velocity dispersion due to beam smearing), showing consistency with the low dispersion inferred by the simple spectroscopic analysis.

As requested by the referee, we are now showing examples of the H α fitting towards the central (most important) region as well as fits to representative spaxels in the red and blue parts of the rotation gradient. Due to editorial concerns, we place this technical information in Supplementary Material where we also showcase corner plots of the posteriors of the individual spaxel fits. The posteriors in Supplemental Material Fig. 3 clearly show that the line is well-resolved near the inferred dynamical center (though due to beam smearing, as revealed by the MOKA3D analysis), while becoming completely unresolved in the outer parts.

We acknowledge the referee's concerns about kinematic errors from individual spaxel fits. The spaxels were fit independently, hence the fitting errors obtained do not include the cross-correlation introduced by beam smearing, which is significant as our target is only resolved across 2 beams. Thus, the fitting errors on individual spaxels are of order of 2 km/s and are lower limits. Hence, formal calculations of the errors of the binned velocity points would give an unrealistic <1 km/s precision. In the original version of the paper, we mitigated this issue by using the maximum error of each binned spaxel as the error on the bin. In the updated manuscript, we utilize a more self-consistent approach in estimating the contribution of correlated uncertainties. In this approach, we initially use the formal errors on each bin and scale them up until the best performing model in the fitted suite gives a reduced $\chi^2 = 1$. This approach yields an upper limit on the errors of each radial bin as it neglects the fact that higher reduced χ^2 values are acceptable and assumes the best performing model in the set is the ground truth. Following this approach, we find that the fitting errors need to be scaled up by a factor of 6, which is in line with the minimum factor of 5 expected from oversampling the native pixel scale of $0.1''$ to $0.02''$. Most importantly, the best performing model in the set is still a Keplerian point mass, which outperforms compact mass distribution by the same significance.

Another important factor to consider is that spectroastrometric data points are, by construction, unaffected by beam smearing. Hence, the uncertainties on them are considerably more robust and they end up driving our result. We provide separate verification of the spectroastrometric approach in the response to the next point raised by the referee.

As for MOKA3D, the code takes fully into account the errors on the individual spaxels, together with the light distribution and beam smearing.

2. Kinematical constraints.

Taking the V values in Fig. 2 at face value, the resolved kinematics alone show no strong radial trend—the blue points are almost consistent with a flat distribution. If the errors are underestimated, this interpretation would be even stronger. Only when combined with the spectroastrometric result at 10–15 pc scales (pink points), the radial trend appears Keplerian.

I find the approach of combining a single spectroastrometric point with resolved kinematics incomplete and somewhat misleading. As the authors note, spectroastrometry provides advantages beyond the nominal spatial resolution, but it can also be applied to larger scales. It is unclear why a full spectroastrometric analysis independent of the resolved kinematics could not be performed. Instead of using one velocity channel ($0 < V < 100$ km/s), the authors could have used several channels to constrain a radial gradient. For example, the velocity channel $0 < V < 20$ – 25 km/s and $25 < V < 50$ should have sufficient signal since it corresponds to the line peak. Such an analysis would provide an independent constraint on gas kinematics and could be compared with the resolved kinematic results. The authors may argue that the S/N is insufficient for more detailed velocity channel analysis, but if that is the case, it raises doubt about the feasibility of the spatially resolved analysis itself.

We first would like to clarify that the observed velocity field in Fig.1 is only apparently flat – the fact that it does not seem to rise steeply towards the centre is due to beam smearing; however, the black hole signature is visually indicated by the sharp change of sign in the centre: without a central point mass the velocity field would be smoothly declining toward the centre. Quantitatively, this aspect is captured by the MOKA3D full fitting.

As for the 1D, binned velocity field, the referee is correct when saying that indeed, the Keplerian nature of the binned velocity field is driven by the spectroastrometry points without which we can not distinguish between a point mass and $R < 10$ pc star cluster; we now point this out explicitly in the text. To test the robustness of the spectroastrometric measurements and their correspondence with the radially binned velocity field, we repeat our analysis with finer splitting of the emission line, as suggested by the referee. However, we note that the width of each spectral channel is 30 - 35 km/s hence we can not split the line as finely as suggested by the referee. Instead, we adopt $0 < V < 30$ km/s and $30 < V < 60$ km/s channels. The spectroastrometric curve constructed this way is shown in the revised Fig. 2 (lighter magenta points) and is entirely consistent with a Keplerian point-mass interpretation despite the large

reduction in S/N in the high velocity channels increasing the measurement errors by a factor of 5. Additionally, to avoid any misunderstanding in the interpretation of Fig.2 we have removed the binned point at <50 pc, which are affected by beam smearing and use only the combination of central spectro-astrometry and binned velocities at large radii (>100 pc), which are not affected by beam smearing.

We would also like to clarify that the use of spectro-astrometry and radial binned channels is to obtain a more visual test relative to the far more advanced, but less visually intuitive MOKA3D full fitting. The two approaches (MOKA3D and spectro-astrometry) are completely independent of each other, and the goal of showing both was to illustrate that they reach consistent results, supporting each other. Yet, if the referee deems the spectroastrometry + radial binning approach confusing, then we are open to entirely removing it.

3. Comparison with a Keplerian curve.

It is misleading to denote the combined velocity and velocity dispersion as V (e.g., in Fig. 2), since readers might confuse it with the line-of-sight velocity. It should instead be expressed differently, for example as $\sigma_{\text{total}} = (V^2 + \sigma^2)^{1/2}$.

It is unclear whether directly comparing σ_{total} to a Keplerian curve is appropriate. While the velocity dispersion should indeed be added to the rotational velocity to represent the total gravitational potential, σ_{total} would not follow a Keplerian profile if the pressure component dominates—which appears to be the case for this LRD.

We would like to clarify that the pressure component does not dominate the kinematics, particularly those near the center: as already discussed above in the centre $V_{\text{rot}} \cdot \sin i \sim 55$ km/s (i.e. about 70 km/s deprojected) and $\sigma \sim 35$ km/s; in the outer parts we only measure upper limits on the velocity dispersion, as discussed in the previous points. Additionally and even more importantly, as already mentioned above, MOKA3D reveals that the apparent, observed velocity dispersion can be fully accounted for by the velocity field convolved with the surface brightness via the beam smearing; further implying that the intrinsic velocity dispersion is sub-dominant. Anyway, as suggested by the referee, in Fig. 2a we are now showing the line-of-sight velocity, (i.e. not $(V^2 + \sigma^2)^{1/2}$), to avoid confusion, and clearly demonstrating that it follows a Keplerian curve when combined with spectroastrometry. This is further strengthened by the consistency of the finer split spectroastrometry points as discussed in the response to the previous point.

As discussed above, the object is not in a configuration where pressure dominates, hence this issue is not critical to our conclusions.

Independently of all of this, the MOKA3D takes into account the full dynamical model, including rotation and any putative pressure support (which MOKA3D finds negligible). Once again, we can remove the spectro-astrometry part if that turns out to be confusing.

4. Possibility of outflows

The IFU maps in Fig. 1 can naturally be interpreted as showing a biconical outflow, though the outflow velocity seems much lower than in typical local Seyfert galaxies with similar Eddington ratios. It is possible that a broad wing (outflow component) of the narrow line is not detected due to the low S/N.

The authors argue that outflows can be ruled out based on velocity channel analysis and MOKA fitting results, but the brief description provided is insufficient to assess this claim. The S/N in the high-velocity (>500 km/s) channels would be very low, as they correspond to the faint wings of the line profile. Given that outflows have wide opening angles and that the astrometric position is affected by emissivity distribution and extinction, Extended Fig. 5 does not convincingly constrain the scenario.

Regarding the MOKA fitting, the method allows flexible input parameters. The MOKA outflow fitting result appears significantly worse than the point-mass or NSC cases, but even the flux distribution in Extended Fig. 6 does not match well, raising the question of whether a better fit might be possible. I do not mean to imply that the authors performed poorly, but rather to emphasize that essential methodological details are missing from this manuscript.

Just to avoid any possible misunderstanding, we recall that the elongated morphology is a consequence of the gravitational lensing shear.

As pointed out by the referee themselves, the velocity is much lower than what seen in outflows driven by similar luminosity AGN.

On the point that a high velocity faint wings associated with a putative outflow would remain undetected, we completely agree with the referee. However, it is not clear why that would affect our fitting of the (well detected) narrow component - the referee's outflow scenario, by construction, should not affect our interpretation, simply because the outflow's wings are not detected.

Regarding the referee's comments about MOKA3D, we totally agree with them that the description of this scenario was too short. We have now expanded the description of the analysis in fitting the observed narrow emission under the hypothesis that the narrow line kinematics are outflow-dominated. We now clarify that the free parameters of the outflow fit are the conical axis inclination with respect to the line of sight (and on the plane of the sky), the intrinsic outflow radial velocity, which is assumed to be constant along the entire outflow, and the position angle of the outflow cone (see Extended Data Table 1). The MOKA3D fitting procedure is aimed at reproducing the observed line profile, under the assumption that the chosen geometry and velocity field are correct. Therefore, although constrained values of inclination and velocity can be determined, if the assumed geometry (conical in this case) and velocity profile (constant radial velocity in this case) are incorrect, the obtained moment maps will show systematic residuals. This is exactly what is happening when fitting the observed kinematic under the assumption of a bi-conical outflow configuration, which is interpreted as incorrect (see Marconcini+23 for more extensive discussion of possible fit degeneracies). As shown in Supplementary Material Fig. 7 the best-fit biconical model leaves systematic residuals which resemble rotation, which points to the fact that a simple outflow with constant radial velocity cannot reproduce the observed features as, by construction, it does not include a rotation pattern which consequently appears in residual maps. In order to better reproduce the velocity field with an outflow scenario one should force a non-physical radial velocity profile of the outflow that mimics a Keplerian rotation, which has never been observed in any known outflow.

Moreover, as the referee has pointed out - it is very unlikely that the observed low velocities could be ascribed to an ionised outflow. In general, ionised outflows in AGN at $\log(L) \sim 44 - 45$ erg/s (like QSO1) are observed to have velocities of at least 400 km/s (see e.g. Fiore+17). As a comparison, surveys in the local and high- z Universe typically consider as ionised outflow signature an emission lines width of $W_{\{80\}} > 500-600$ km/s, which is considerably higher than our case (Ruschel-Dutra+2021, Kakkad+20, Wylezalek+20, Tadhunter+18, Harrison+14).

Based on these assessments, I conclude that the authors should present a more detailed analysis in a full-length paper to convincingly demonstrate their results to the community.

We understand the referees concerns. However, the importance and potential impact of the result is the reason we are submitting to Nature. Its multidisciplinary nature means that the summary of the result will reach an audience wider than the astrophysics community, while the Methods and Supplemental sections will provide ample space for

technical information of interest. In addition, we plan to publish the MOKA3D code together with the spectroastrometry pipeline, both of which have been improved thanks to the referee's comments, upon the acceptance of the paper.

Referee #2 (Remarks to the Author):

This is an important observational result, with deep consequences for understanding little red dots, as the authors discuss. There are only a couple of points I wanted some clarification on.

1) The spectroastrometric (SA) method measures the flux-weighted photocenter as a function of velocity channel, not the location of the gas producing a given velocity. A single point derived from SA is an effective, model-dependent (emissivity, geometry, inclination, etc) radius. Is the radius uncertainty for the SA point reflecting all of these issues?

Indeed, the SA radius, as well as the rest of the radii in the 1D rotation curve do not automatically correspond to physical radii and are subject to an inclination correction, which is however a systematic offset affecting all velocity points. The simplified SA analysis is unable to constrain inclination or disk geometry and inclination, and this is why the simple SA method results in a lower limit to the black hole mass. We rely on MOKA3D for tighter constraints and for assessing the disc inclination. Aside from the disc inclination, SA is not really model-dependent, it is simply a very empirical determination of the position shift as a function of velocity channel. The referee is absolutely correct that SA relies on a photocentre that is emissivity-weighted; yet, we are interested in relative displacement of the velocity channels and therefore this is not a major issue. However, once again, MOKA3D fully takes into account the emissivity distribution in its modelling, hence overcoming any potential uncertainty on this regard that might be faced in the SA approach. MOKA3D, with its refined approach, gives results fully consistent with the simple SA approach.

2) What is the impact on the results if the SA point is not considered? If the rotation curve was measured firmly within the black hole sphere of influence, as the authors claim, then presumably the overall estimates would not hinge so strongly on the innermost point. It would be useful to repeat the fit without the SA constraint and report how the BH mass, inclination, and enclosed mass profile change. This will make the sensitivity on the SA point explicit rather than implicit.

In principle yes, one could recover the Keplerian rotation by simply using the radially binned velocities, without SA, but in that case one would have to convolve the fitting

Keplerian rotation curve with the (emissivity-weighted) beam smearing - this is essentially what MOKA3D does (though in the more extensive 3D space); therefore, undertaking this approach in 1D would not be an independent (nor improved) test, so it would not be very useful. Instead, the advantage of SA is that it is independent of the beam smearing, while it relies on the S/N of each spectral channel, so this provides an independent test of the velocity field.

Additionally, please note that, following the request of Referee #1, we have now replaced (in Fig.2) the inner binned velocity fields, with the SA in finer bins.

It should be noted that MOKA3D, which takes into account the entirety of the flux distribution, is independent of spectroastrometry and is not reliant on a single measured point, but rather the entirety of the observed flux and velocity distribution, and provides fully consistent (and tighter) results.

3) The authors use σ_N as a proxy for stellar velocity dispersion to estimate the black hole sphere of influence as 270 pc. Notice that this is considerably larger than typical values (e.g., $\sigma_* = 150$ km/s gives ~ 8 pc for the quoted BH mass). What is the justification for using σ_N instead of σ_* ?

We do not have σ_* , as stellar features are not detected. Yet, the velocity dispersion of the narrow emission lines has been found to trace the stellar velocity dispersion reasonably well, with Bezanson+ 2018 finding the two offset by 0.1-0.15 dex (at $z=1$, the only cross-check available so far at high- z). Hence, given that our narrow line dispersion is ~ 22 km/s, we would not expect the stellar σ to exceed 31 km/s. Either way, the sphere of influence is only a preliminary (and rough) estimate used to motivate the study presented in the paper - we do not use this information a priori in any of our analysis and in any of our conclusions. We can remove the initial reference to the expected sphere of influence, if the referee thinks it is causing confusion.

Referee #3 (Remarks to the Author):

The paper "A direct black hole mass measurement in a Little Red Dot at the Epoch of Reionization" reports the dynamical detection of a supermassive black hole of 10^7 suns in a triply imaged source at $z=7.04$. If the measurements are sound, then this would be a truly remarkable and very important result. To our knowledge at this moment, the only dynamical black hole mass measurements outside the Local Volume have been made in either very luminous quasars with GRAVITY or in a couple of lensed galaxies at Cosmic Noon, with

billion solar mass black holes. This measurement would be even more remarkable as the nature of the Little Red Dots is still very much under debate. This paper infers a system that is completely black hole dominated, a truly remarkable result if true.

However, we feel that the paper should present both (a) more examination of the data, and possible systematic uncertainties in the emission line fitting, (b) a stronger set of tests on the fidelity of the spectro-astrometry, and finally (c) a much clearer exposition of the MOKA-3D fitting, including a full accounting of free parameters.

Major Point 1: Broad+Medium+Absorption+Narrow-line fitting. There are many components fitted to the Ha line. Since the BH detection hinges on the velocity field of the narrow H α line, we would like to see:

– an investigation into the degeneracy between the fit to the absorber, the two broad components, and the resulting flux, velocity, and dispersion field of the narrow line. We appreciate that the broad components are unresolved, but apparently the absorber sits in front of the broad line (according to D'Eugenio+2025) and so we are not convinced that fixing the absorber in velocity and width across the cube is necessarily justified.

We acknowledge that the description of the fitting was not detailed enough in the previous version. In the Supplementary Material we now show fits to three representative spaxels - one coincident with the dynamical center and two 150pc away in the blueshifted and redshifted direction. As shown there, no significant residuals are left by any of the fits. This rules out significant spatial variation of the absorber properties as the spaxels shown are 1 PSF width apart. Hence our choice of fixing the absorber properties is justified. This is also expected on simple physics grounds: Balmer absorption is due to very dense gas (e.g. Juodžbalis+24, Inayoshi & Maiolino+25, Ji+25) and very likely located in a compact nuclear region, probably the outer parts of the BLR or a dense and thick pseudo-atmosphere around the nuclear region; additionally, the absorption is against the continuum and the broad component, which are both unresolved, hence also the absorption should be unresolved too.

In addition to the best-fit plots, we are now showing the corner plots of the posteriors of fits to both the full line profile and the BLR-subtracted spaxel spectra. The posteriors show no significant degeneracy between the inferred narrow line properties and the scaling of the fixed BLR model. Hence, the two components of the line are well decoupled by the spectral resolution.

– an exploration of the outcome of allowing multiple absorbers, or a varying velocity and width.

We appreciate that a single absorber with a narrow line super-imposed can also be modeled as two separate absorbers on either side of the line. Hence our absorption feature may indeed have a multi-component structure. However, multi-component absorber models are inherently degenerate, thus we prefer a simpler single-absorber model, which leaves no significant residuals, as a detailed exploration of the absorption features and their nature is beyond the scope of this work.

As shown in Supplemental Material, fixing the width and velocity of the absorption feature does not leave significant distance-dependent residuals, hence leaving these parameters free would likely lead to overfitting.

– propagating forward the uncertainty from these fits into the resulting velocity measurements

Uncertainties on the narrow line profile are already estimated by propagating the uncertainties of the full line profile fitting into the data cube used by MOKA3D. For what concerns spectroastrometric analysis, the dominant uncertainty is the rms derived from signal-free regions of each channel image.

Major Point 2: The spectro-astrometry seems like a brilliant way to improve the resolution in this case. However, we are not clear on how the error bars shown in Extended Data Fig. 1 are translated into the rotation curve presented in the main text. Please explore:

– the significance of the spatial offset in more detail. For instance, Abuter+2024 (GRAVITY) present a null test to prove that they have measured an actual spatial offset as a function of velocity. This would be a worthwhile type of test to perform here.

We thank the referee for this very useful suggestion. We thus perform a null test of a similar type to Abuter+24, utilizing JWST IFU calibration observations of stars showing that our procedure produces no noticeable spatial offsets for point-sources. We provide detailed descriptions and results of the procedure in the Supplementary Material.

– the sensitivity of the spatial offset to the exact spectral division between red and blue?

This is essentially a point very similar to a point raised also by Referee 1 - splitting the narrow component in more velocity channels, and illustrated in the revised Fig.2, also tackles this concern - although in this case the S/N is lower, the result remains consistent with the original analysis.

– propagating the uncertainty in the absorption line fit into the uncertainty in the total narrow line profile

As already discussed above, uncertainties on the narrow line profile are already estimated by propagating the uncertainties of the full line profile fitting into the data cube used by MOKA3D.

Major Point 3: From the information presented, it is very hard to understand both (a) what free parameters are involved in the MOKA-3D fit or (b) what about the velocity field causes such a strong preference for an unresolved point mass.

- Please plot the models as well as the residuals for the fits.
- Please provide a complete list of the free parameters constrained by MOKA-3D, and how well-constrained they are.
- Of specific concern, please describe whether the dispersion is a free parameter, and whether it is allowed to vary spatially. Typically in black hole mass determination papers based on gas velocity fields, there is significant attention paid to possible turbulent (non-gravitational) support.
- Please describe the physical model generating the flux distribution in each case. Given that the observed flux distribution is asymmetric, we do not fully understand why the rotating disk model does such a clean job with the flux distribution. Similarly, the dispersion field is not symmetric, which seems unusual for a rotating disk. Please explain how the MOKA-3D model fits the dispersion field so cleanly.
- It would be useful here to comment on the spatial resolution achieved in the maps fitted by MOKA-3D. It seems that the sphere of influence is not reached, so how are the constraints on M_{bh} so tight?

We thank the reviewers for raising these points which helped us to improve the presentation of our kinematics analysis, which underpins the manuscript. In the following text we address each point.

- **We have now included the MOKA3D Keplerian model beneath the observed maps in Fig.1, and moved the associated residuals (previously in Fig.2b) to Fig.1 (while the MOKA3D residuals for the NSC have been moved to the Methods). The data and the model are visually identical as the deviations are very small and appreciated only in the residuals, this is why we had initially opted for showing only residual maps in the previous version of the manuscript. Yet, following the referees' suggestions, we have updated Fig. 1 to show the best fit model along with data and residuals.**

- **The complete list of the MOKA3D free parameters with their uncertainties are now listed clearly in the Extended Tab.1 and specified in the relevant sections of the text. For a comprehensive description of the model uncertainty determination we refer to the second published paper on the MOKA3D code (Marconcini+25) where a complete overview is presented.**
- **The observed features of the data cube were well reproduced without including any additional velocity dispersion parameters. MOKA3D modeling works by inferring the best-fit parameters reproducing the observed line profile. The flux is then recovered by a flux-weighting procedure (we refer to Marconcini+25 for details). Hence, MOKA3D will reproduce the data as long as the assumed geometry is correct. The reason for this has to be ascribed to how the model works. In particular, once the model infer the best-fit parameters that allow to reproduce the observed line percentiles, then the entire line profile is recovered by the weighting procedure, which will adapt the intrinsic flux of each cloud in the model to the observed flux, thus matching the model and observed spectra, if the assumed geometrical configuration is correct. Therefore, introducing additional parameters would be redundant and introduce unnecessary degeneracies.**
- **The main advantage of MOKA3D is that there is no intrinsic assumption on the flux distribution within the fictitious distribution of clouds of the 3D model. As now emphasised in the manuscript, the model can reproduce the observed features only if the assumed geometry and velocity fields allow for clouds to be located in correct 3D positions. Operationally, the velocity dispersion map tells us about the flux-weighted line broadening, which the correct model will be able to reproduce on a spaxel-by-spaxel basis, thanks to the weighting procedure.**
- **The tightness of the constraint on the BH is the natural consequence of how MOKA3D works. In particular, as now clarified in the text, MOKA3D is designed to reproduce the observed line profile in the data-cube under the assumption of a specific geometrical and kinematical configuration, i.e. in this case a disc with rotation pattern dictated by a point mass. MOKA3D creates a data cube, with the same spatial and spectral binning as the observed cube, and populates it with a distribution of fictitious clouds with an intrinsic kinematic profile determined by the Keplerian velocity field with the BH mass left as free parameter. For each BH mass we generate a model cube, assign the clouds the corresponding kinematic profile**

(depending on M_{BH}), project the model onto the sky and verify how well the observed line profile is reproduced by the model line profile. Once the best fit BH mass is inferred, we create the single best-fit model cube and weight the model clouds based on the observed flux in each 3D pixel (i.e. spatial pixel and spectral channel, see Marconcini+23 for details on this procedure). The capability of the best-fit model in reproducing the observed 2D features in the observed moment maps will then only depend on whether the assumed kinematics and geometry are correct and a best-fit parameter can be recovered.

We find the text very clear and well-written, but do feel we need more clarity on the methods and uncertainties as outlined above to understand the robustness of the measurement.

We thank the referees for dedicating their time to provide additional comments, the responses to which are provided in bold below.

The changes in the manuscript (Main text, Methods, and Supplementary Material) are marked in bold.

Due to the Editor requesting a shortened Methods section, we have moved the sections describing the technical details of narrow line fitting and spectroastrometry to the Supplementary Information (these are not marked in bold, as they are not changed, but simply moved). This has, however, necessitated some amendments to the Methods text to keep the flow of the manuscript. Those changes are highlighted in bold.

Referee #1 (Remarks to the Author):

I thank the authors for carefully addressing my comments and suggestions in their revision of the manuscript and in the response letter. Although I still believe that a full analysis in a regular journal would be preferable given the complexity of the analysis and the limitations of the data, I recommend the manuscript for publication and leave the assessment of the results to the readers.

On Page 8 after Eq.1, there is a typo: "ar unresolved" should be "are unresolved".

Thank you for spotting this. This has been fixed.

We are grateful to the referee for the several comments, which have improved the manuscript.

Referee #2 (Remarks to the Author):

Regarding my third comment about using the narrow line velocity dispersion. For the sake of completeness please explicitly mention in the supplementary material that no stellar features are detected and given the that σ_N tracks the σ_{star} (quoting the appropriate reference), they end up with the estimated BH sphere of influence estimate.

This has been clarified in the Supplementary Material.

Otherwise, I am happy with the author's response.

We are grateful to the referee for their help and insights during the review process.

Referee 3/4

Point 1: Spectro-astrometry

The authors have suggested that their results hinge more on the MOKA-3D modeling than on the spectro-astrometry. If so, perhaps this comment can be disregarded, but we would appreciate hearing the thoughts of the authors on this new concern about the spectro-astrometry. We studied Stern, Hennawi, & Pott (2015), a detailed exploration of spectro-astrometry techniques for luminous quasars. They have a subsection 3.4.1 on contamination from the narrow-line region, that is gas that is more spatially extended than the broad-line region. In this subsection, they explain that if there is NLR emission on the spatial scale of r_{PSF} , then it will be very hard to filter out this emission. Because the position at a given frequency is r -weighted (see their Eq. 12), even a very small fraction of the photons emerging from such large scales will dominate the position measurement at that frequency.

Please address whether it is possible to measure sub-PSF positions using narrow-line gas, when there is demonstrably extended emission on the scale of the PSF.

The referees are correct in pointing out that for a generic AGN host the flux from larger scales could drown out a signal from the BLR, which is what Stern, Hennawi & Pott were attempting to resolve. However, unlike Stern et al., in our study we are not attempting to spectroastrometrically map out the BLR, but rather focus on the NLR dynamics itself (that is why we subtract the broad line emission from the cube). Therefore, in our case all emission originates from similar spatial scales and we do not have the issue of the ~ 3 dex difference between the BLR and NLR radii, which is what drives the contamination in Stern, Hennawi & Pott (2015). Therefore, standard spectroastrometric considerations apply and it is possible to measure sub PSF scales as per Iverson (2007).

Point 2: Velocity Field

If the MOKA-3D modeling is the critical measurement, then we find it even more important that the authors investigate realistic uncertainties in the narrow H α velocity. We are convinced by the authors that it is fair to fix the absorber parameters from the spatially averaged fit. However, there is uncertainty in the velocity and width of both the absorber and the broad emission line fits that is covariant with the uncertainty in the narrow velocity field. Please take into account the uncertainty in the absorber and broad-line fits when fitting the narrow line (e.g., by sampling from the posterior for the absorber and broad-line fit).

Having fixed the properties of the BLR and absorber (which the referees agree with), the only remaining free parameter is the normalization, and we have illustrated in Supplementary Material Figs. 2 and 3 that this parameter is not degenerate with the velocity field $v(x, y)$ of the narrow component - this was already mentioned in

Supplementary Material, but we have clarified it further. Therefore, the marginalized posterior distribution on the velocity field $v(x, y)$, hence the associated uncertainties, are already included and taken into account by the MOKA3D analysis. Regardless – precisely because the BLR+absorber model is spatially fixed – even if we repeated the analysis with a slightly different BLR+absorber model (essentially, a different realization of the BLR+absorber within the measurement uncertainties), this would produce a spatially uniform systemic shift in $v(x, y)$; such a systematic uncertainty would be subtracted from the velocity field and interpreted as a different systemic redshift of the source.

Point 3: MOKA-3D Fitting

We thank the authors for adding more detail about the MOKA-3D fitting, but we still find ourselves a bit confused. The claim seems to be that a velocity gradient of ~ 10 km/s measured over a spatial scale of ~ 300 pc over two independent beams is sufficient to rule out a spatially extended distribution of mass. Is that in fact the claim? It seems impossible on the face of it. The inferred mass is $\sim 10^7$ Msun. 100-200 pc seems consistent with mass-size measurements for galaxies of this mass. This leads to our first question: could the authors rule out an extended (~ 150 pc-scale) disk with $M^* \sim 10^7$ Msun?

We thank the referees for the opportunity to clarify this point further. The simple exponential disc is totally excluded on multiple grounds. On the 1D plot of Fig.2 the exponential disc would smoothly decline to zero toward the centre, contrary to what is observed. Below we show what the rotation curve would be for an exponential disc with the same parameters suggested by the referees ($\sim 10^7$ Msun 100-200 pc effective radius), both for the 1D plot of Fig.2 and in the case of the 2D velocity field fitted with MOKA3D (see attached figure below).

As shown in the above plot, the exponential disc model with $R_e=150$ pc and $M = 10^7 M_{\odot}$ does reproduce the velocity field at >100 pc, motivating our attempts to estimate extended mass contribution to the total dynamical mass. On the other hand, a pure exponential disc fails completely at reproducing the centrally peaked velocity profile producing an overall reduced $\chi^2 = 22$, an order of magnitude worse than concentrated mass models and the best-fit Keplerian curve.

The equivalent plot for MOKA3D is shown below and clearly showcases considerably worse residuals than the Keplerian model shown in Fig. 1 of the main text.

These aspects have been clarified in the Methods, including the two additional plots above.

In terms of the presented fits, we are still confused about the following points:

(1) In the primary fit (Fig 1), why does the Keplerian point mass model have a flux asymmetry? Furthermore, can the authors explain how the Keplerian model reproduces the velocity dispersion map so precisely given the spatial resolution? In particular, there are a few disconnected pixels with $\sigma=40 \text{ km/s}$ on either side of the dynamical center. How does the model reproduce that pattern so closely after being beam-smeared? Perhaps it would help to show the unsmoothed (true) velocity and velocity dispersion distributions?

The intrinsic Keplerian model presented in Fig.1 does not intrinsically have an asymmetric flux distribution. Indeed, as shown in the image attached below of the *unweighted* Keplerian model not PSF-convolved, each cloud is assigned a constant weight (i.e., flux), which translates to a uniform flux distribution of the model. The key feature of MOKA3D is that it can then reproduce the velocity field and flux distribution in cases where the gas emission throughout the disc is not uniform (which is typically the case in most galaxies, especially at high redshift). More specifically, the asymmetric flux distribution is recovered thanks to two features, which represent some of the main advantages of MOKA3D:

- 1- The fact that the Keplerian model in consideration has emitting model clouds which perfectly match, given the geometric configuration and orientation, the observed velocities covered by the narrow Ha emission line.
2. The cloud-weighting procedure. As explained in the manuscript, once the best-fit 3D model is inferred and the unweighted model is created (see e.g., the attached figure), each model cloud is assigned a precise flux, matching directly the flux measured from the data.

Therefore, combining points 1 and 2 allows reproducing asymmetric/clumpy/non-uniform flux distribution for any geometry, if the parameters used to create the model are correct (see also Fig.5 in Marconcini+23, which describes exactly this effect). Finally, we stress that the capability of MOKA3D in reproducing non uniform flux distributions with analytic velocity distributions is among the major strengths of the model, as deeply discussed in the presentation paper (Marconcini+23) and in section “*MOKA3D kinematics modeling*” of the Method section of the current manuscript, demonstrating that complex features observed in moment maps can be explained by non uniform intrinsic flux distributions rather than complicated velocity fields.

These points were already discussed in the Methods; however, we have expanded a little bit, especially by including the figure above showing the intrinsic, unweighted underlying kinematical model.

(2) A related question is why the flux distributions (which are not shown) and residuals for the other models do NOT match the observed asymmetry in flux. If a symmetric rotating Keplerian disk can have an asymmetric flux distribution, why not these other models?

This is a direct consequence of the previous point and happens if either one or both of points 1-2 described above fail. In the case of the NSC model the flux distribution, as well as the first and second moment maps, are not reproduced as properly as the Keplerian model due to the absence of model clouds at the correct velocities (i.e., at velocity bins where the observed data show non-zero emission). Therefore, it is impossible for the model to reproduce such emission due to the absence of model clouds that could gain the necessary emission to reproduce the observed features. Similarly, the Plummer sphere

model, despite performing better than the NSC model, still shows larger residuals compared to the pure Keplerian model, still reproducing well the asymmetric emission observed.

(3) If the authors adopt more realistic velocity errors as suggested in Point 2, and fix the (arbitrary?) flux distributions to match what is observed, then are the spatially extended models presented here actually inconsistent with the data?

The crucial feature for MOKA3D is not the amount of flux in each velocity bin (which is equally redistributed among the model clouds in the same bin) but rather the line width, as it affects the inferred best-fit parameters. We agree with the referees that the uncertainties of the broad Balmer component and the absorber will propagate on the subtracted narrow H α component, but such uncertainties will impact only the flux in each velocity bin of the narrow component, leaving the line width unchanged, therefore not affecting our kinematic results. Anyway, as discussed above, these uncertainties are fully incorporated and taken into account by the MOKA3D analysis.